# LeanAgent: Lifelong Learning for Formal Theorem Proving

**Adarsh Kumarappan**[*,1]**, Mo Tiwari**[*,2]**, Peiyang Song**[1]**, Robert Joseph George**[1]**,
Chaowei Xiao**[3]**, Anima Anandkumar**[1]
[1]California Institute of Technology, [2]Stanford University, [3]University of Wisconsin, Madison
{adarsh, psong, rgeorge, anima}@caltech.edu, motiwari@stanford.edu, cxiao34@wisc.edu

## Abstract

Large Language Models (LLMs) have been successful in mathematical reasoning tasks such as formal theorem proving when integrated with interactive proof assistants like Lean. Existing approaches involve training or fine-tuning an LLM on a specific dataset to perform well on particular domains, such as undergraduate-level mathematics. These methods struggle with generalizability to advanced mathematics. A fundamental limitation is that these approaches operate on static domains, failing to capture how mathematicians often work across multiple domains and projects simultaneously or cyclically. We present LeanAgent, a novel lifelong learning framework for formal theorem proving that continuously generalizes to and improves on ever-expanding mathematical knowledge without forgetting previously learned knowledge. LeanAgent introduces several key innovations, including a curriculum learning strategy that optimizes the learning trajectory in terms of mathematical difficulty, a dynamic database for efficient management of evolving mathematical knowledge, and progressive training to balance stability and plasticity. LeanAgent successfully generates formal proofs for 155 theorems across 23 diverse Lean repositories where formal proofs were previously missing, many from advanced mathematics. It performs significantly better than the static LLM baseline, proving challenging theorems in domains like abstract algebra and algebraic topology while showcasing a clear progression of learning from basic concepts to advanced topics. In addition, we analyze LeanAgent's superior performance on key lifelong learning metrics. LeanAgent achieves exceptional scores in stability and backward transfer, where learning new tasks improves performance on previously learned tasks. This emphasizes LeanAgent's continuous generalizability and improvement, explaining its superior theorem-proving performance.

## 1 Introduction

Mathematics can be expressed in informal and formal languages. Informal mathematics utilizes natural language and intuitive reasoning, whereas formal mathematics employs symbolic logic to construct machine-verifiable proofs (Kevin Buzzard, 2019). State-of-the-art large language models (LLMs), such as o1 (OpenAI, 2024) and Claude (Claude Team, 2024), produce incorrect informal proofs (Zhou et al., 2024). This highlights the importance of formal mathematics in ensuring proof correctness and reliability. Interactive theorem provers (ITPs), such as Lean (De Moura et al., 2015), have emerged as tools for formalizing and verifying mathematical proofs. However, constructing formal proofs using ITPs is complex and time-consuming; it requires extremely detailed proof steps and involves working with extensive mathematical libraries.

Recent research has explored using LLMs to generate proof steps or complete proofs. For example, LeanDojo (Yang et al., 2023) introduced the first open-source framework to spur such research. Existing approaches typically involve training or fine-tuning LLMs on a specific dataset (Jiang et al., 2022). However, data scarcity in formal theorem proving (Polu et al., 2022) hinders the generalizability of these approaches (Liu et al., 2023). For example, ReProver, the retrieval-augmented

---

[*]Equal contribution.

LLM from the LeanDojo family, uses a retriever fine-tuned on Lean's math library, `mathlib4` (mathlib4 Community, 2024). Although `mathlib4` contains over 100,000 formalized mathematical theorems and definitions, it covers primarily up to undergraduate mathematics. Consequently, ReProver performs poorly on more challenging mathematics, such as Terence Tao's formalization of the Polynomial Freiman-Ruzsa (PFR) Conjecture (Tao et al., 2024).

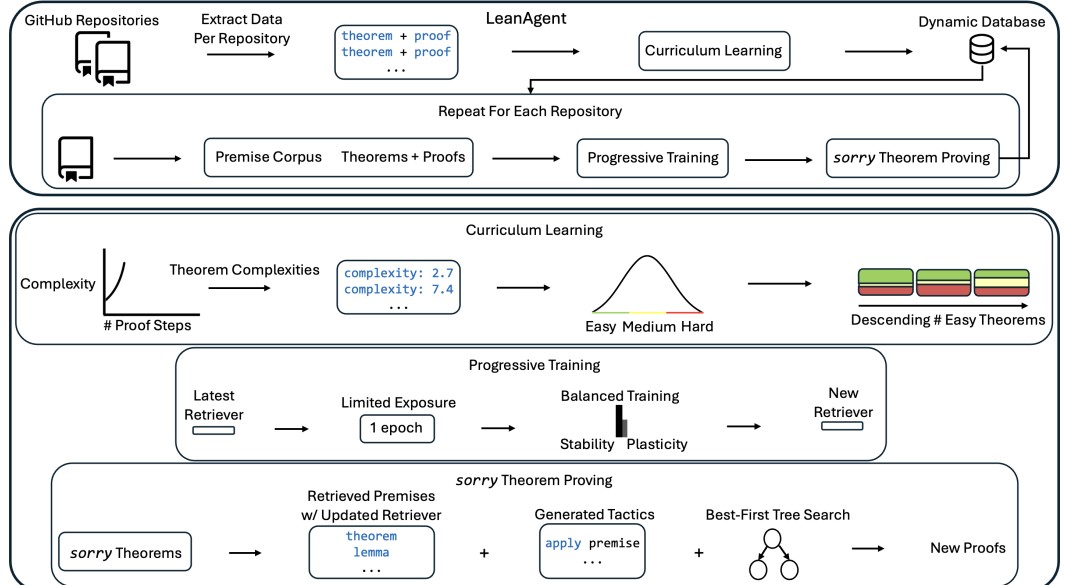

Figure 1: LeanAgent overview. LeanAgent searches for Lean repositories and uses LeanDojo to extract theorems and proofs. It uses curriculum learning, computing theorem complexity as $e^S$ ($S$ = proof steps) and calculating the 33rd and 67th complexity percentiles across all theorems to sort repositories by easy theorem count. LeanAgent adds the curriculum to its dynamic database. As a retrieval-based framework, LeanAgent generates a dataset (premise corpus and a collection of theorems and proofs) for each repository in the curriculum and progressively trains its retriever. Progressive training happens over one epoch to prevent forgetting old knowledge. Then, LeanAgent uses the updated retriever in a search-based method to generate formal proofs for theorems where formal proofs were previously missing, known as *sorry* theorems. It adds new proofs to the database.

The dynamic nature of mathematical research exacerbates this generalizability issue. Mathematicians often formalize across multiple domains and projects simultaneously or cyclically. For example, Terence Tao has worked on various projects in parallel, including formalizations of the PFR Conjecture, symmetric mean of real numbers, classical Newton inequality, and asymptotic analysis (Tao; 2024a; Tao et al., 2024; Tao, 2024b). Patrick Massot has been formalizing Scholze's condensed mathematics and the Perfectoid Spaces project (Community, 2024a;b). These examples highlight a critical gap in current theorem-proving AI approaches: the lack of a system that can adapt and improve across multiple, diverse mathematical domains over time, given limited Lean data availability.

**Connection to Lifelong Learning.** Crucially, how mathematicians formalize is relevant to lifelong learning, i.e. learning multiple tasks without forgetting (Wang et al., 2024b). A significant challenge is catastrophic forgetting: when adaptation to new distributions leads to a loss of understanding of old ones (Jiang et al., 2024). The core challenge is balancing plasticity (the ability to learn and adapt) with stability (the ability to retain existing knowledge) (Wang et al., 2024b). Increasing plasticity to learn new tasks efficiently can lead to overwriting previously learned information. However, enhancing stability to preserve old knowledge may impair the model's ability to acquire new skills (van de Ven et al., 2024). Achieving the right balance is key to continuous generalizability in theorem proving.

**LeanAgent.** We present LeanAgent, a novel lifelong learning framework for theorem proving. As shown in Figure 1, LeanAgent's workflow consists of (1) deriving complexity measures of theorems

to compute a curriculum for learning, (2) progressive training to learn while balancing stability and plasticity, and (3) searching for proofs of *sorry* theorems by leveraging a best-first tree search, all while using a dynamic database to manage its evolving mathematical knowledge. LeanAgent works with any LLM; we implement it with retrieval for improved generalizability (Yang et al., 2023).

We employ a simple progressive training method to avoid catastrophic forgetting. Progressive training allows LeanAgent to continuously adapt to new mathematical knowledge while preserving previously learned information. This process involves incrementally training the retriever on newly generated datasets from each repository in the curriculum. Starting with a pre-trained retriever (e.g., ReProver's retriever based on ByT5 (Xue et al., 2022)), LeanAgent trains on each new dataset for one additional epoch. Restricting progressive training to one epoch helps balance stability and plasticity. Crucially, this training is repeated for each dataset generated from the database, gradually expanding LeanAgent's knowledge base. This approach increases the space of possible proof states (where a state consists of a theorem's hypotheses and current proof progress) while adding new premises to the premise embeddings. More sophisticated lifelong learning methods like Elastic Weight Consolidation (EWC) (Kirkpatrick et al., 2017), which uses the Fisher Information Matrix to constrain important weights for previous tasks, result in excessive plasticity. The uncontrolled plasticity is due to the inability of these methods to adapt parameter importance as theorem complexity increases. This forces rapid changes in parameters crucial for learning advanced concepts. Such methods fail to adapt to the evolving complexity of mathematical theorems, making them unsuitable for lifelong learning in theorem proving.

Extensive experiments across 23 diverse Lean repositories demonstrate LeanAgent's advancements in lifelong learning for theorem proving. LeanAgent successfully generates formal proofs for 155 theorems across these 23 repositories where formal proofs were previously missing, known as *sorry* theorems, many from advanced mathematics. For example, it proves challenging *sorry* theorems in abstract algebra and algebraic topology related to Coxeter systems and the Hairy Ball Theorem (Coxeter, 2024; Hairy Ball Theorem, 2024). LeanAgent also proves 7 theorems using exploits found within Lean's type system. We find that LeanAgent demonstrates progressive learning in theorem proving, initially proving basic *sorry* theorems and significantly advancing to more complex ones. It significantly outperforms the static ReProver baseline in terms of proving new *sorry* theorems. We have issued pull requests to the respective repositories with the newly proven *sorry* theorems. Some of these proofs utilized unintended constructs within the repositories' implementations, which are currently being addressed through appropriate fixes.

In theorem proving, we find that stability, without losing too much plasticity, is crucial for continuous generalizability to new repositories. Backward transfer (BWT), where learning new tasks improves performance on previously learned tasks, is essential in theorem proving (Wang et al., 2024b). Mathematicians require a lifelong learning framework for theorem proving that is both *continuously generalizable* and *continuously improving*. We conduct an extensive ablation study using six lifelong metrics carefully proposed or selected from the literature. LeanAgent's simple components of curriculum learning and progressive training improve stability and BWT scores substantially, emphasizing its continuous generalizability and improvement and explaining its superior *sorry* theorem proving performance.

## 2 PRELIMINARIES

**Neural Theorem Proving.** The current state-of-the-art of learning-based provers employs Transformer-based (Vaswani et al., 2017) LLMs that process expressions as plain text strings (Azerbayev et al., 2024; Xin et al., 2024b; Shao et al., 2024). In addition, researchers have explored complementary aspects like proof search algorithms (Lample et al.; Wang et al., 2023). Moreover, other works break the theorem-proving process into smaller proving tasks (Song et al., 2024; Wang et al., 2024a; Lin et al., 2024).

**Premise Selection.** A critical challenge in theorem proving is the effective selection of relevant premises (Irving et al., 2016; Tworkowski et al.). However, many existing approaches treat premise selection as an isolated problem (Wang & Deng, 2020; Piotrowski et al., 2023) or use selected premises only as input to symbolic provers (Alama et al., 2014; Mikuła et al., 2024).

**Retrieval-Augmented LLMs.** While retrieval-augmented language models have been extensively studied in areas like code generation (Lu et al., 2022; Zhou et al., 2023), their application to formal theorem proving is relatively new. However, relevant architectures have been researched in natural language processing (NLP) (Lu et al., 2024; Borgeaud et al., 2022; Thakur et al., 2024).

**Lifelong Learning.** Lifelong learning addresses catastrophic forgetting in sequential task learning (Chen et al., 2024). Approaches include regularization methods (Kirkpatrick et al., 2017), memory-based techniques (Lopez-Paz & Ranzato, 2017; Chaudhry et al., 2019; Shin et al., 2017), and knowledge distillation (Li & Hoiem, 2017; Kim et al., 2023). Other strategies involve dynamic architecture adjustment (Mendez & Eaton, 2021) and recent work on gradient manipulation and selective re-initialization (Chen et al., 2024; Dohare et al., 2024). We justify not using these strategies in Appendix A.6.

**Curriculum Learning in Theorem Proving.** Prior work created a synthetic inequality generator to produce a curriculum of statements of increasing difficulty (Polu et al., 2022). For reinforcement learning, an existing work used the length of proofs to help determine rewards (Zombori et al., 2019).

## 3 METHODOLOGY

A useful lifelong learning strategy for theorem proving requires (a) a repository order strategy and (b) a learning strategy. We solve (a) with curriculum learning to utilize the structure of Lean proofs and (b) with progressive training to balance stability and plasticity. LeanAgent consists of four main components: curriculum learning, dynamic database management, progressive training of the retriever, and *sorry* theorem proving. Further methodology details are in Appendix A.1 and a discussion of why curriculum learning works in theorem proving is available in Appendix A.6.

### 3.1 CURRICULUM LEARNING

LeanAgent uses curriculum learning to learn on increasingly complex mathematical repositories. This process optimizes LeanAgent's learning trajectory, allowing it to build upon foundational knowledge before tackling more advanced concepts.

First, we automatically search for and clone Lean repositories from GitHub. We use LeanDojo for each repository to extract fine-grained information about their theorems, proofs, and dependencies. Then, we calculate the complexity of each theorem using $e^S$, where $S$ represents the number of proof steps. However, *sorry* theorems, which have no proofs, are assigned infinite complexity. We use an exponential scaling to address the combinatorial explosion of possible proof paths as the length of the proof increases. Further justification for considering this complexity metric is in Appendix A.6.

We compute the 33rd and 67th percentiles of complexity across all theorems in all repositories. Using these percentiles, we categorize non-*sorry* theorems into three groups: easy (theorems with complexity below the 33rd percentile), medium (theorems with complexity between the 33rd and 67th percentiles), and hard (theorems with complexity above the 67th percentile). We then sort repositories by the number of easy theorems they contain. This sorting forms the basis of our curriculum, with LeanAgent starting on repositories with the highest number of easy theorems.

### 3.2 DYNAMIC DATABASE MANAGEMENT

Then, we add the sorted repositories to LeanAgent's custom dynamic database using the data LeanAgent extracted. This way, we can keep track of and interact with the knowledge that LeanAgent is aware of and the proofs it has produced. We also include the complexity of each theorem computed in the previous step into the dynamic database, allowing for efficient reuse of repositories in a future curriculum. Details of the database's contents and features can be found in Appendix A.1.

For each repository in the curriculum, LeanAgent uses the dynamic database to generate a dataset by following the same procedure used to make LeanDojo Benchmark 4 (details in Appendix A.1). This dataset includes a collection of theorems and their proofs. Each step of these proofs contains detailed annotations, such as how the step changes the state of the proof. A state consists of a theorem's hypotheses and the current progress in proving the theorem. As such, this pairing of

theorems and proofs demonstrates how to use specific tactics (functions) and premises in sequence to prove a theorem. In addition, the dataset includes a premise corpus, serving as a library of facts and definitions.

### 3.3 PROGRESSIVE TRAINING OF THE RETRIEVER

LeanAgent then progressively trains its retriever on the newly generated dataset. This strategy allows LeanAgent to continuously adapt to new mathematical knowledge from the premises in new datasets while preserving previously learned information, crucial for lifelong learning in theorem proving. Progressive training achieves this by incrementally incorporating new knowledge from each repository.

Although LeanAgent works with any LLM, we provide a specific implementation here. We start with ReProver's retriever, a fine-tuned version of the ByT5 encoder (Xue et al., 2022), leveraging its general pre-trained knowledge from `mathlib4`. We train LeanAgent on the new dataset for an additional epoch. This limited exposure helps prevent overfitting to the new data while allowing LeanAgent to learn essential new information. LeanAgent's retriever, and therefore the embeddings it generates, are continuously updated during progressive training. Thus, at the end of the current progressive training run, we precompute embeddings for all premises in the corpus generated by LeanAgent's current state to ensure that we properly evaluate LeanAgent's validation performance. To understand how LeanAgent balances stability and plasticity, we save the model iteration with the highest validation recall for the top ten retrieved premises (R@10). This is a raw plasticity value: it can be used to compute other metrics that describe LeanAgent's ability to adapt to and handle new types of mathematics in the latest repository (details in Sec. 4). Then, we compute the average test R@10 over all previous datasets the model has progressively trained on, a raw stability value.

As mentioned previously, we repeat this procedure for each dataset we generate from the database, hence the progressive nature of this training. Progressive training adds new premises to the premise embeddings and increases the space of possible proof states. This allows LeanAgent to explore more diverse paths to prove theorems, discovering new proofs that it couldn't produce with its original knowledge base.

### 3.4 *sorry* THEOREM PROVING

For each *sorry* theorem, LeanAgent generates a proof with a best-first tree search by generating tactic candidates at each step, in line with prior work (Yang et al., 2023). Using the embeddings from the entire corpus of premises we previously collected, LeanAgent retrieves relevant premises from the premise corpus based on their similarity to the current proof state, represented as a context embedding. Then, it filters the results using a corpus dependency graph to ensure that we only consider premises accessible from the current file. We add these retrieved premises to the current state and generate tactic candidates using beam search. Then, we run each tactic candidate through Lean to obtain potential next states. Each successful tactic application adds a new edge to the proof search tree. We choose the tactic with the maximum cumulative log probability of the tactics leading to it. If the search reaches a dead-end, we backtrack and explore alternative paths. We repeat the above steps until the search finds a proof, exhausts all possibilities, or reaches the time limit of 10 minutes.

If LeanAgent finds a proof, it adds it to the dynamic database. The newly added premises from this proof will be included in a future premise corpus involving the current repository. Moreover, LeanAgent can learn from the new proof during progressive training in the future, aiding further improvements.

## 4 EXPERIMENTS

### 4.1 EXPERIMENTAL SETUP

We devote this section to describing two types of experiments: (1) *sorry* Theorem Proving: We compare *sorry* theorem proving performance between LeanAgent and ReProver. We also examine the progression of LeanAgent's proven *sorry* theorems during lifelong learning to the end of life-

Table 1: Selected repository descriptions

| Repository | Description |
|---|---|
| PFR | Polynomial Freiman-Ruzsa Conjecture |
| Hairy Ball Theorem | Algebraic topology result |
| Coxeter | Coxeter groups |
| Mathematics in Lean Source | Lean files for the textbook |
| Formal Book | Proofs from *THE BOOK* |
| MiniF2F | Math olympiad-style problem solving |
| SciLean | Scientific computing |
| Carleson | Carleson's Theorem |
| Lean4 PDL | Propositional Dynamic Logic |

Table 2: Accuracy in proving *sorry* theorems across repositories. Accuracy is calculated as (proven theorems / total *sorry* theorems). "LA" denotes LeanAgent and "RePprover+" denotes the setting where we update RePprover on *all* 23 repositories at once. "During" shows accuracy during lifelong learning, "Add. After" shows *additional* accuracy after lifelong learning, and "Total" shows the combined accuracy. "MIL" stands for Mathematics in Lean Source and "Hairy Ball" refers to the Hairy Ball Theorem repository. Repositories with no *sorry* theorems or no proven ones are not shown. The best accuracy for each repository is in bold. As noted previously, we progressively train on MiniF2F after the initial curriculum to demonstrate the use case of formalizing in a new repository after learning a curriculum. As such, we don't evaluate LeanAgent after lifelong learning on MiniF2F.

| Repository | #*sorry*s | LA Accuracy (%) | | | RePprover Accuracy (%) | RePprover+ Accuracy (%) |
|---|---|---|---|---|---|---|
| | | Total | During | Add. After | | |
| MIL | 29 | **72.4** | 48.3 | 24.1 | 48.3 | 55.2 |
| MiniF2F | 406 | **24.4** | 24.4 | - | 20.9 | 20.9 |
| Formal Book | 29 | **10.3** | 6.9 | 3.4 | 6.9 | **10.3** |
| SciLean | 294 | **9.2** | 7.5 | 1.7 | 8.2 | 8.5 |
| Hairy Ball | 14 | **7.1** | 0.0 | 7.1 | 0.0 | **7.1** |
| Coxeter | 15 | **6.7** | 6.7 | 0.0 | 0.0 | **6.7** |
| Carleson | 24 | **4.2** | 4.2 | 0.0 | **4.2** | **4.2** |
| Lean4 PDL | 30 | **3.3** | 3.3 | 0.0 | **3.3** | **3.3** |
| PFR | 37 | **2.7** | 2.7 | 0.0 | 0.0 | 0.0 |

long learning. This shows LeanAgent's continuous generalizability and improvement. (2) Lifelong Learning Analysis: We conduct an ablation study with six lifelong learning metrics to explain LeanAgent's superiority in *sorry* theorem proving. Moreover, these results explain LeanAgent's superior handling of the stability-plasticity tradeoff. Please see Appendix A.2 for experiment implementation details. We release LeanAgent at https://github.com/lean-dojo/LeanAgent.

**Repositories.** We evaluate our approach on a diverse set of 23 Lean repositories to assess its generalizability across different mathematical domains (Skřivan, 2024; Kontorovich, 2024; Avigad, 2024; Tao et al., 2024; Renshaw, 2024; Fermat's Last Theorem, 2024; DeepMind, 2024; Carneiro, 2024; Wieser, 2024; Mizuno, 2024; Murphy, 2024; Formal Logic, 2024; Con-nf, 2024; Gadgil, 2024; Yang, 2024; Zeta 3 Irrational, 2024; Firsching, Moritz, 2024; Monnerjahn, 2024; van Doorn, 2024; Dillies, 2024; Hairy Ball Theorem, 2024; Coxeter, 2024; Gattinger, 2024). Details of key repositories are in Table 1. Further details of these repositories, including commits and how we chose the initial curriculum and the sub-curriculum (described in Sec. 4.2), are in Appendix A.3.

## 4.2 *sorry* THEOREM PROVING

We compare the number of *sorry* theorems LeanAgent can prove, both during and after lifelong learning, to the RePprover baseline. We use RePprover as the baseline because we use its retriever as LeanAgent's initial retriever in our experiments.

In addition, a use case of LeanAgent is proving in a new repository after learning a curriculum; we progressively train on MiniF2F to demonstrate this. Note that we choose the Lean4 version

**PFR**

```
lemma condRho_of_translate {Ω S : Type*} [MeasureSpace
Ω] (X : Ω → G) (Y : Ω → S) (A : Finset G) (s:G) :
condRho (fun ω ↦ X ω + s) Y A = condRho X Y A := by
  simp only [condRho, rho_of_translate]
```

**SciLean**

```
theorem re_float (a : Float)
: RCLike.re a = a := by
  exact RCLike.re_eq_self_of_le le_rfl
```

**Coxeter**

```
lemma invmap.of_eq {S:Set G} [CoxeterSystem G S] {s :S}
: invmap S s = s := by
  simp [CoxeterSystem.Presentation.invmap]
  unfold CoxeterSystem.toMatrix
  apply CoxeterSystem.monoidLift.mapLift.of
```

**MiniF2F**

```
theorem induction_12dvd4expnp1p20
(n : ℕ) :
12 | 4^(n+1) + 20 := by
  norm_num
  induction' n with n hn
  simp
  omega

theorem amc12a_2002_p6
(n : ℕ)
(h₀ : 0 < n) :
∃ m, (m > n ∧ ∃ p, m * p ≤ m + p) := by
  lift n to ℕ+ using h₀
  cases' n with n
  exact ⟨_, lt_add_of_pos_right _
zero_lt_one, 1, by simp)
```

**Formal Book**

```
theorem wedderburn (h: Fintype R): IsField R
:= by
  apply Field.toIsField
```

Figure 2: Case studies of LeanAgent's new proofs. LeanAgent shows an ability to work with these repositories, often able to retrieve the necessary premises (highlighted). For example, LeanAgent proves a *sorry* theorem from PFR, condRho_of_translate, by simply expanding definitions, showing its proving ability on the PFR repository. In addition, LeanAgent could prove re_float from SciLean during lifelong learning, while ReProver could not. Moreover, its MiniF2F proofs demonstrate its ability to generate relatively longer and more complex proofs for complex mathematics. LeanAgent's proof of invmap.of_eq and wedderburn represents its theorem proving capabilities with abstract algebra premises.

of the MiniF2F repository (Yang, 2024) and disregard its separation into validation and test splits (reasoning in Appendix A.5). LeanAgent's success rate on the Lean4 version of the MiniF2F test set is also in Appendix A.5. Moreover, a mathematician could use LeanAgent for (1) an initial curriculum $A$, and later (2) a sub-curriculum $B$. LeanAgent can then help the mathematician prove in the repositories in curriculum $A + B$. To demonstrate this scenario, we continue LeanAgent on a sub-curriculum $B$ of 8 repositories.

Results are in Table 2, with case studies in Figure 2. Appendix A.5 provides a more thorough discussion, including an ablation study, and contains the complete theorems and proofs relevant to this section.

LeanAgent demonstrates continuous generalizability and improvement in theorem-proving capabilities across multiple repositories. LeanAgent's proofs are a superset of the *sorry* theorems proved by ReProver in most cases. Moreover, to isolate the effect of curriculum learning, we compare LeanAgent against ReProver+, the ReProver model updated on all 23 repositories at once, and notice that LeanAgent outperforms it on several repositories, emphasizing the importance of curriculum learning. Overall, LeanAgent progresses from basic concepts (arithmetic, simple algebra) to advanced topics (abstract algebra, topology).

**PFR.** LeanAgent can prove a *sorry* theorem from this repository, while ReProver cannot. It also generalizes to a different commit (not included in progressive training), uncovering 7 system exploits. LeanAgent proves two theorems with just the `rfl` tactic, one of which ReProver cannot, and proves 5 *sorry* theorems with a $0 = 1$ placeholder theorem statement.

**SciLean.** During lifelong learning, LeanAgent proves theorems related to fundamental algebraic structures, linear and affine maps, and measure theory basics. By the end of lifelong learning, it proves concepts in advanced function spaces, sophisticated bijections, and abstract algebraic structures.

**Mathematics in Lean Source.** During lifelong learning, LeanAgent proves theorems about basic algebraic structures and fundamental arithmetic properties. By the end of lifelong learning, it proves more complex theorems involving quantifier manipulation, set theory, and relations.

**MiniF2F.** ReProver demonstrates proficiency in basic arithmetic, elementary algebra, and simple calculus. However, by the end of lifelong learning, LeanAgent handles theorems with advanced

number theory, sophisticated algebra, complex calculus and analysis, abstract algebra, and complex induction.

**Sub-curriculum.** In the Formal Book repository, LeanAgent progresses from proving basic real analysis and number theory theorems to more advanced abstract algebra, exemplified by its proof of Wedderburn's Little Theorem. For the Coxeter repository, LeanAgent proves a complex lemma about Coxeter systems, showcasing its increased understanding of group theory. In the Hairy Ball Theorem repository, LeanAgent proves a key step of the theorem, demonstrating improved performance in algebraic topology. Only LeanAgent can prove these theorems, demonstrating that it has much more advanced theorem-proving capabilities than ReProver.

## 4.3 LIFELONG LEARNING ANALYSIS

To our knowledge, no other lifelong learning frameworks for theorem proving exist in the literature. As such, we conduct an ablation study with six lifelong learning metrics to showcase LeanAgent's superior handling of the stability-plasticity tradeoff. These results help explain LeanAgent's superiority in *sorry* theorem proving performance. We compute these metrics for the original curriculum of 14 repositories.

Specifically, the ablation study consists of seven additional setups constructed from a combination of learning and dataset options. Options for learning setups are progressive training with or without EWC. Dataset setups involve a dataset order and construction. Options for dataset orders involve Single Repository or Merge All, where each dataset consists of all previous repositories and the new one. Given the most popular repositories on GitHub by star count, options for dataset construction include popularity order or curriculum order. Appendix A.3 shows these orders and additional repository details.

**Metrics.** We use six lifelong learning metrics: Windowed-Forgetting 5 (WF5), Forgetting Measure (FM), Catastrophic Forgetting Resilience (CFR), Expanded Backward Transfer (EBWT), Windowed-Plasticity 5 (WP5), and Incremental Plasticity (IP). A description of these metrics is in Table 3 (De Lange et al., 2023; Wang et al., 2024b; Díaz-Rodríguez et al., 2018). Our reasoning for considering these metrics is detailed in Appendix A.4.

Table 3: Description of lifelong learning metrics.

| Metric | Description | Target | Type |
| --- | --- | --- | --- |
| WF5 | Measures forgetting over a 5-task window | Lower | Existing |
| FM | Average performance drop on old tasks | Lower | Existing |
| CFR | Ratio of min to max average test R@10 | Higher | Proposed |
| EBWT | Average improvement on old tasks after learning new ones | Higher | Existing |
| WP5 | Max average test R@10 increase over a 5-task window | Higher | Existing |
| IP | Rate of validation R@10 change per task | Higher | Proposed |

We describe why we introduce two new metrics to address specific aspects of lifelong learning in theorem proving:

- **Catastrophic Forgetting Resilience (CFR).** This metric captures LeanAgent's ability to maintain performance on its weakest task relative to its best performance, crucial in the presence of diverse mathematical domains.

- **Incremental Plasticity (IP).** IP provides a more granular view of plasticity than aggregate measures and is sensitive to the order of tasks, particularly relevant in lifelong learning for theorem proving.

In addition, these metrics in the Merge All strategy measure cumulative knowledge refinement rather than isolated task performance (details in Appendix A.4). Due to these interpretational differences, we analyze Single Repository and Merge All setups separately. We consider an improvement of at least 3% to be significant.

**Single Repository Analysis.** We first analyze the Single Repository results from Table 4. LeanAgent demonstrates superior stability across multiple metrics. The WF5 metric is 75.34% lower for

Table 4: Comparison of lifelong learning metrics across setups. The best scores for each metric are in bold.

| Metric | Single Repository | | | | Merge All | | | |
|---|---|---|---|---|---|---|---|---|
| | **LeanAgent** | **Setup 1** | **Setup 2** | **Setup 3** | **Setup 4** | **Setup 5** | **Setup 6** | **Setup 7** |
| WF5 ($\downarrow$) | **0.18** | 7.60 | 7.17 | 0.73 | 15.83 | **2.23** | 13.34 | 5.82 |
| FM ($\downarrow$) | **0.85** | 6.53 | 4.04 | 2.11 | 10.50 | 4.06 | 11.44 | **3.80** |
| CFR ($\uparrow$) | **0.88** | 0.87 | 0.88 | 0.85 | 0.76 | **0.94** | 0.75 | 0.90 |
| EBWT ($\uparrow$) | **1.21** | 0.51 | 1.04 | 0.76 | -0.20 | **0.73** | -1.34 | -0.39 |
| WP5 ($\uparrow$) | 2.47 | 0.89 | 1.47 | **3.42** | 0.00 | 0.09 | 0.00 | **0.11** |
| IP ($\uparrow$) | 1.02 | 0.36 | 0.26 | **1.06** | -1.50 | **-0.64** | -1.71 | -0.89 |
| **Legend** | | | | | | | | |

| Single Repository: | Merge All: |
|---|---|
| Setup 1: No EWC, Popularity Order | Setup 4: No EWC, Popularity Order |
| Setup 2: EWC, Popularity Order | Setup 5: No EWC, Curriculum Learning |
| Setup 3: EWC, Curriculum Learning | Setup 6: EWC, Popularity Order |
| | Setup 7: EWC, Curriculum Learning |

LeanAgent than the next best setup, suggesting it maintains performance over a window more effectively. Its FM score is 59.97% lower than Setup 3's, showcasing its resilience against catastrophic forgetting. Furthermore, LeanAgent, Setup 1, and Setup 2 demonstrate high and consistent resilience against catastrophic forgetting, with CFR values above 0.87 and minimal ($\pm 0.01$) differences. This underscores LeanAgent's ability to continuously generalize over time. In addition, LeanAgent has a 16.25% higher EBWT, indicating its ability to continuously improve over time.

In contrast, Setup 3 exhibits characteristics of higher plasticity. It shows a 38.26% higher WP5 over LeanAgent, indicating a greater ability to rapidly adapt to new tasks in a window. This is complemented by its 3.98% higher IP over LeanAgent, suggesting a more pronounced improvement on new tasks over time. However, these plasticity gains come at a significant cost: Setup 3 suffers from more severe catastrophic forgetting, as evidenced by its significantly worse stability metrics compared to LeanAgent. This excessive plasticity in Setup 3 stems from EWC's inability to adapt parameter importance as theorem complexity increases. EWC preserves parameters important for simpler theorems, which may not be crucial for more complex ones. Consequently, these preserved parameters resist change while other parameters change rapidly for complex theorems. This forces the model to become more plastic overall, relying heavily on non-preserved parameters for new, complex theorems.

LeanAgent's favorable stability and EBWT scores make it the most suitable for lifelong learning in the Single Repository setting.

**Merge All Analysis.** Next, we analyze the Merge All setups from Table 4. Setup 5's WF5 metric is 61.68% lower than the next best setup (Setup 7), suggesting Setup 5 balances and retains knowledge across an expanding dataset most effectively. Furthermore, Setup 5's CFR score is 3.77% higher than that of Setup 7, again demonstrating high and consistent resilience in the face of an expanding, potentially more complex dataset. However, Setup 7 has a 6.44% lower FM score than Setup 5's, showcasing its ability to maintain performance on earlier data points. Moreover, Setup 5 is the only setup with a positive EBWT, indicating that learning new tasks improves performance on the entire historical dataset. The other setups have a negative EBWT, indicating performance degradation on earlier tasks after learning new ones.

Only Setups 5 and 7 have a non-zero WP5, suggesting the ability to adapt to the growing complexity of the combined dataset. The zero values for Setups 4 and 6 indicate that popularity order struggles to show improvement when dealing with merged data. However, although Setup 5 has the highest IP score with a 27.75% improvement over Setup 7, all 4 setups have negative IP values. This indicates a decrease in validation R@10 over time, suggesting that the Merge All strategy struggles to maintain performance.

Experiment 5's favorable stability and EBWT scores suggest it is the best at balancing the retention of earlier knowledge with the adaptation to new data in a combined dataset. However, its negative IP value indicates a fundamental issue with its approach.

**Comparative Analysis and Insights.** Although the metrics have different interpretations in the Single Repository and Merge All settings, we can still draw some meaningful comparisons by focusing on overall trends and relative performance. We must consider that the negative IP values in Merge All setups indicate a significant issue. This drawback outweighs the potential benefits seen in other metrics like WP5, as it indicates a fundamental inability to maintain and improve performance in a continuously growing dataset. In contrast, LeanAgent demonstrates a positive IP, indicating its ability to incorporate new knowledge. This, combined with its superior stability and EBWT metrics relative to other Single Repository methods, suggests that LeanAgent is better suited than Setup 5 for continuous generalizability and improvement.

**Consistency with *sorry* Theorem Proving Performance.** This lifelong learning analysis is consistent with LeanAgent's *sorry* theorem proving performance. LeanAgent's superior stability metrics (WF5, FM, and CFR) explain its ability to maintain performance across diverse mathematical domains, as evidenced by its success in proving theorems from various repositories like SciLean, Mathematics in Lean Source, and PFR. Its high EBWT score aligns with its progression from basic concepts to advanced topics in theorem proving. While LeanAgent shows slightly lower plasticity (WP5 and IP) compared to some setups, this trade-off results in better overall performance, as reflected in its ability to prove a superset of *sorry* theorems compared to ReProver in most cases. This analysis demonstrates LeanAgent's overall superiority in lifelong learning for theorem proving.

## 5 CONCLUSION

We have presented LeanAgent, a lifelong learning framework for theorem proving that achieves continuous generalizability and improvement across diverse mathematical domains. Key components include a curriculum learning strategy, progressive training approach, and custom dynamic database infrastructure. LeanAgent successfully generates formal proofs for 155 theorems where formal proofs were previously missing and uncovers 7 exploits across 23 Lean repositories, including from challenging mathematics. This highlights its potential to assist in formalizing complex proofs across multiple domains and identifying system exploits. For example, LeanAgent successfully proves challenging theorems in abstract algebra and algebraic topology. It outperforms the ReProver baseline in proving new *sorry* theorems, progressively learning from basic to complex mathematical concepts. Moreover, LeanAgent shows significant performance in forgetting measures and backward transfer, explaining its continuous generalizability and continuous improvement.

Future work could explore integration with Lean Copilot, providing real-time assistance with a mathematician's repositories. In addition, a limitation of LeanAgent is its inability to prove certain theorems due to a lack of data on specific topics, such as `odeSolve.arg_x₀.semiAdjoint_rule` in SciLean about ODEs. To solve this problem, future work could use reinforcement learning for synthetic data generation during curriculum construction. Moreover, future work could use LeanAgent with additional math LLMs and search strategies.

### ACKNOWLEDGMENTS

Adarsh Kumarappan is supported by the Summer Undergraduate Research Fellowships (SURF) program at Caltech. Anima Anandkumar is supported by the Bren named chair professorship, Schmidt AI 2050 senior fellowship, and ONR (MURI grant N00014-18-12624). We thank Terence Tao for detailed discussions and feedback that significantly improved this paper. We thank Zulip chat members for engaging in clarifying conversations that were incorporated into the paper.

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

# A  APPENDIX

## A.1  FURTHER METHODOLOGY DETAILS

**Repository Scanning and Data Extraction.** We use the GitHub API to query for Lean repositories based on sorting parameters (e.g., by repository stars or most recently updated repositories). We maintain a list of known repositories to avoid; the list can be updated to allow LeanAgent to re-analyze the same repository on a new commit or Lean version.

We clone each identified repository locally using the Git version control system. To ensure compatibility with our theorem-proving pipeline, we check the Lean version required by each repository and compare it with the supported versions of our system. If the required version is incompatible, we skip the repository and move on to the next one. Otherwise, LeanAgent switches its Lean version to match the repository's version. This version checking is performed by parsing the repository's configuration files and extracting the specified Lean version.

**Dynamic Database Management.** This database contains many key features that are useful in our setting. For example, it can add new repositories, update existing ones, and generate merged datasets from multiple repositories with customizable splitting strategies. In addition, it can query specific theorems or premises across repositories, track the progress of proof attempts (including the proof status of *sorry* theorems), and analyze the structure and content of Lean proofs, including tactic sequences and proof states.

The database keeps track of various details: Repository metadata; theorems categorized as already proven, *sorry* theorems that are proven, or *sorry* theorems that are unproven; premise files with their imports and individual premises; traced files for tracking which files have been processed; detailed theorem information, including file path, start/end positions, and full statements; and traced tactics with annotated versions, including the proof state before and after application.

If we encounter duplicate theorems between repositories while merging repositories, we use the theorem from the repository most recently added to the database. We deduplicate premise files and traced files by choosing the first one encountered while merging the repositories. We also generate metadata containing details of all the repositories used to generate the dataset and statistics regarding the theorems, premise files, and traced files in the dataset, such as the total number of theorems.

We provide the user with many options to generate a dataset. To generate the set of theorems and proofs, the default option is to simply use the theorems, proofs, premise files, and traced files from the current curriculum repository in the database. Specifically, we use the random split from LeanDojo to create training, validation, and testing sets. We refrain from using the novel split from LeanDojo, as we would like LeanAgent to learn as much as possible from a repository to perform well on its hardest theorems. The data in the splits include details about the proofs of theorems, including the URL and commit of the source repository, the file path of the theorem, the full name of the theorem, the theorem statement, the start and end positions in the source file, and a list of traced tactics with annotations. The validation and test split each contain 2% of the total theorem and proofs, following the methodology from LeanDojo. Moreover, the database uses a topological sort over the traced files in the repository to generate the premise corpus. This corpus is a JSON Lines file, where each line is a JSON object consisting of a path to a Lean source file, the file's imports, and the file's premise statements and definitions.

**Progressive Training of the Retriever.** We describe some additional steps for progressive training. To precompute the embeddings, we use a single forward pass with batch processing to serialize and tokenize premises from the entire corpus. Then, we use the retriever's encoder to process the batches and generate embeddings.

*sorry* **Theorem Proving.** We start by processing the premise corpus to use it more efficiently during premise retrieval. This involves initializing a directed dependency graph to represent each file path in the corpus, adding files as nodes and imports as edges, and creating a transitive closure of this graph. We also track all premises encountered during this process, building a comprehensive knowledge base.

Crucially, we limit retrieval to a subset of all available premises to aid the effectiveness of the results. Specifically, we choose the top 25% of accessible and relevant premises, following ReProver's method.

**Proof Integration and Pull Request Generation.** We integrate the generated proofs into the original Lean files and create pull requests to propose the changes to the repository owners. This aids the development of these repositories and functions as more training data for future research.

To achieve this, in a temporary Git branch, we iterate over the Lean files and locate the *sorry* keywords corresponding to the generated proofs. We then replace these *sorry* keywords with the actual proof text, working from the bottom of each file upward to preserve the position of theorems. After integrating the proofs, we commit our changes, push them, and create a pull request for the repository on GitHub.

## A.2 EXPERIMENT IMPLEMENTATION DETAILS

We use ReProver's retriever trained on the random split from LeanDojo. We use four NVIDIA A100 GPUs with 80GB of memory each for progressive training. LeanAgent uses a distributed architecture leveraging PyTorch Lightning and Ray for parallel processing. We use bfloat16 mixed precision and optimize with AdamW (Loshchilov & Hutter, 2017) with an effective batch size of 16 (achieved through a batch size of 4 with gradient accumulation over 4 steps). In the first 1,000 steps, the learning rate warms up linearly from 0 to the maximum value of $10^{-3}$. Then it decays to 0 using a cosine schedule. In addition, we apply gradient clipping with a value of 1.0. Just as ReProver does during training, we sample 3 negative premises per example, including 1 in-file negative premise. The maximum sequence length for the retriever is set to 1024 tokens. The maximum sequence length for the generator is set to 512 tokens for input and 128 tokens for output.

The prover uses a best-first search strategy with no limit on the maximum number of expansions of the search tree. It generates 64 tactic candidates and retrieves 100 premises for each proof state. LeanAgent uses ReProver's tactic generator for the experiments. We generate tactics with a beam search of size 5. We used 4 CPU workers, 1 per GPU. Due to the wide variety of repositories and

experimental setups that we tested, the time for each experiment widely varied. For example, the experiments in Table 10 took from 4 to 9 days to complete.

Furthermore, we do not compare LeanAgent with any existing LLM-based prover besides ReProver because LeanAgent is a framework, not a model. As mentioned previously, it can be used with any LLM. As such, a comparison would be impractical for reasons including differences in data, pre-training, and fine-tuning. We only compare with ReProver because we use ReProver's retriever as the starting one in LeanAgent, allowing for a more faithful comparison.

Moreover, we do not compare with Aesop because it is not an ML model. We aim to improve upon ML research for theorem proving, such as ReProver. Moreover, Aesop is not a framework, but ReProver was included as the starting point of the LeanAgent framework, which is why we compare LeanAgent to ReProver. Rather than comparing against existing tools, we aim to understand how lifelong learning can work in theorem proving.

Furthermore, although LeanAgent can work with other LLMs such as Llemma (Azerbayev et al., 2024) and DeepSeek-Prover (Xin et al., 2024a), using these LLMs in our work would require architectural modifications that go beyond the scope of our current work. For example, the 7B model DeepSeek-Prover as well as the 7B and 34B Llemma models are not retrieval-based. As such, rather than progressively training a retriever, we would progressively train the entire model. This may be feasible with methods such as Gradient Low-Rank Projection (Zhao et al.), but this would lead to fundamentally different usage than we currently demonstrate. Specifically, rather than using a best-first tree search approach as we do with ReProver's retriever and tactic generator, we may instead need to generate the entire proof at once. This setup is quite dissimilar from our current evaluation, and so these results may be too dissimilar from our current evaluation framework.

In addition, we would like to note that because LeanAgent does not claim to contribute a new search algorithm, it can be used with other search strategies such as Hypertree Proof Search (Lample et al.). However, the source code for Hypertree Proof Search was only recently released on GitHub, explaining why we did not use it thus far.

Moreover, the objective function for Elastic Weight Consolidation (EWC) is given by:

$$L(\theta) = L_B(\theta) + \frac{\lambda}{2} \sum_i F_i (\theta_i - \theta_{A,i})^2$$

where $L_B(\theta)$ is the loss for the current task B, $i$ is the label for each parameter, $\theta_{A,i}$ are the parameters from the previous task A, $F_i$ is the Fisher information matrix, and $\lambda$ is a hyperparameter that controls the strength of the EWC penalty. For the setups that use EWC, we performed a grid search over $\lambda$ values in {0.01, 0.1, 1, 10, 100}. For each value, we ran Setup 2 on separate testing repositories. We found 0.1 to yield the best overall stability and plasticity scores.

## A.3 REPOSITORY DETAILS

Table 5: Additional repository descriptions

| Repository | Description |
| --- | --- |
| Prime Number Theorem And | Prime Number Theorem proof |
| Compfiles | Catalog of Olympiad-style math problems |
| FLT | Fermat's Last Theorem proof |
| Debate | Stochastic double-efficient debate protocol |
| Lean4Lean | Implementation of Lean4 kernel in Lean4 |
| Matrix Cookbook | The Matrix Cookbook lemmas |
| Math Workshop | Detailed Lean tutorial |
| LeanEuclid | Euclidean Geometry |
| Foundation | Formal logic results |
| Con-nf | Consistency of Quine's New Foundations |
| Saturn | SAT solver-prover implementation |
| Zeta 3 Irrational | Proof of $\zeta(3)$ irrationality |
| Formalization of Constructable Numbers | Ancient construction problems |
| LeanAPAP | Kelley-Meka bound on Roth numbers |

Table 6: Repository commits. Formalization of Const. Numbers denotes Formalization of Constructable Numbers.

| Repository | Commit |
|---|---|
| PFR | fa398a5b853c7e94e3294c45e50c6aee013a2687 |
| Hairy Ball Theorem | a778826d19c8a7ddf1d26beeea628c45450612e6 |
| Coxeter | 96af8aee7943ca8685ed1b00cc83a559ea389a97 |
| Mathematics in Lean Source | 5297e0fb051367c48c0a084411853a576389ecf5 |
| Formal Book | 6fbe8c2985008c0bfb30050750a71b90388ad3a3 |
| MiniF2F | 9e445f5435407f014b88b44a98436d50dd7abd00 |
| SciLean | 22d53b2f4e3db2a172e71da6eb9c916e62655744 |
| Carleson | bec7808b907190882fa1fa54ce749af297c6cf37 |
| Lean4 PDL | c7f649fe3c4891cf1a01c120e82ebc5f6199856e |
| Prime Number Theorem And | 29baddd685660b5fedd7bd67f9916ae24253d566 |
| Compfiles | f99bf6f2928d47dd1a445b414b3a723c2665f091 |
| FLT | b208a302cdcbfadce33d8165f0b054bfa17e2147 |
| Debate | 7fb39251b705797ee54e08c96177fabd29a5b5a3 |
| Lean4Lean | 05b1f4a68c5facea96a5ee51c6a56fef21276e0f |
| Matrix Cookbook | f15a149d321ac99ff9b9c024b58e7882f564669f |
| Math Workshop | 5acd4b933d47fd6c1032798a6046c1baf261445d |
| LeanEuclid | f1912c3090eb82820575758efc31e40b9db86bb8 |
| Foundation | d5fe5d057a90a0703a745cdc318a1b6621490c21 |
| Con-nf | 00bdc85ba7d486a9e544a0806a1018dd06fa3856 |
| Saturn | 3811a9dd46cdfd5fa0c0c1896720c28d2ec4a42a |
| Zeta 3 Irrational | 914712200e463cfc97fe37e929d518dd58806a38 |
| Formalization of Const. Numbers | 01ef1f22a04f2ba8081c5fb29413f515a0e52878 |
| LeanAPAP | 951c660a8d7ba8e39f906fdf657674a984effa8b |

Table 7: Repository orders (initial curriculum). Note that Popularity Order is by star count on August 20, 2024.

| # | Curriculum Order | Popularity Order |
|---|---|---|
| 1 | Compfiles | SciLean |
| 2 | Mathematics in Lean Source | FLT |
| 3 | Prime Number Theorem And | PFR |
| 4 | Math Workshop | Prime Number Theorem And |
| 5 | FLT | Compfiles |
| 6 | PFR | Debate |
| 7 | SciLean | Mathematics in Lean Source |
| 8 | Debate | Lean4Lean |
| 9 | Matrix Cookbook | Matrix Cookbook |
| 10 | Con-nf | Math Workshop |
| 11 | Foundation | LeanEuclid |
| 12 | Saturn | Foundation |
| 13 | LeanEuclid | Con-nf |
| 14 | Lean4Lean | Saturn |

Table 8: Curriculum order (sub-curriculum)

| # | Curriculum Order |
|---|---|
| 1 | Zeta 3 Irrational |
| 2 | Formal Book |
| 3 | Formalization of Constructable Numbers |
| 4 | Carleson |
| 5 | LeanAPAP |
| 6 | Hairy Ball Theorem |
| 7 | Coxeter |
| 8 | Lean4 PDL |

Table 9: Repository theorem and premise counts

| Repository | Total Theorems | Total Premises |
|---|---|---|
| PFR | 74306 | 109855 |
| Hairy Ball Theorem | 73026 | 131217 |
| Coxeter | 71273 | 127608 |
| Mathematics in Lean Source | 78886 | 117699 |
| Formal Book | 74654 | 112458 |
| MiniF2F | 71313 | 127202 |
| SciLean | 72244 | 129711 |
| Carleson | 73851 | 109334 |
| Lean4 PDL | 20599 | 46400 |
| Prime Number Theorem And | 79147 | 115751 |
| Compfiles | 121391 | 178108 |
| FLT | 75082 | 114830 |
| Debate | 68853 | 103684 |
| Lean4Lean | 2559 | 22689 |
| Matrix Cookbook | 67585 | 102294 |
| Math Workshop | 76942 | 115458 |
| LeanEuclid | 15423 | 40555 |
| Foundation | 25047 | 57964 |
| Con-nf | 29489 | 64177 |
| Saturn | 10982 | 34497 |
| Zeta 3 Irrational | 120174 | 176332 |
| Formalization of Const. Numbers | 74050 | 109645 |
| LeanAPAP | 71090 | 109477 |

**Repository Statistics and Information.** Additional repository descriptions are in Table 5. The commits we used for experiments are in Table 6. Moreover, the repository orders are detailed in Table 7 and Table 8. Furthermore, the total number of theorems and premises per repository (including dependencies) are in Table 9.

**Repository Selection Process.** Many repositories have issues such as incompatibilities with LeanDojo, unsupported Lean versions, and build failures. As such, our process for choosing the 14 repositories in the first curriculum was simply using LeanAgent to extract information from the most popular repositories on GitHub. We disregard incompatible and inapplicable ones, such as those with no theorems. We performed this process on August 20, 2024. We performed a similar process for the eight repositories in the second curriculum with two differences: (1) We checked that the number of *sorry* theorems visible from GitHub was at least 10. This narrowed down the available repositories significantly. (2) We included some more recently updated repositories to provide some variety in the age of the repositories in our curriculum. We performed this process on September 14, 2024. However, many of the repositories that passed this test had fewer than 10 *sorry* theorems when processed by LeanDojo; this is mainly due to the functionalities of LeanDojo.

## A.4 LIFELONG LEARNING METRIC DETAILS

Prior work has noted that lifelong learning methods generally lack standard evaluation metrics (De Lange et al., 2023; Díaz-Rodríguez et al., 2018). As such, our selection primarily focused on metrics that emphasized a change over time, aligning with our problem setup. In addition, we removed metrics that were redundant. For example, prior work suggests that evaluating lifelong learning frameworks only after each task, rather than over time, leads to substantial forgetting (De Lange et al., 2023). As such, we adopt WF and WP in our analysis of LeanAgent. We use a window size of 5 for WF and WP as this represents a relatively medium-term understanding, given that we have 14 repositories. This would provide a balanced interpretation of forgetting and plasticity. Furthermore, we use the EBWT metric, in line with previous work, to evaluate LeanAgent throughout its lifetime rather than simply at the end (Díaz-Rodríguez et al., 2018). Moreover, we chose not to include the Forward Transfer metric as prior work has shown that a lower FM leads to better forward transfer (Chen et al., 2023). As such, we only check FM. We also chose not to include lifelong learning met-

rics for overall performance, such as Time Weighted Cumulative Performance (TWCP), Area Under the Learning Curve (AULC), and Average Accuracy (AA), as these would lead to redundancy in our analysis. Specifically, the metrics we chose were all computed using validation R@10 and the average test R@10, which are already measures of LeanAgent's performance.

We provide some additional details on the metrics we used. Windowed-Forgetting 5 (WF5) quantifies model stability by measuring the maximum performance decrease in average test R@10 over a sliding window of 5 evaluations. Following prior work, we define WF for a given window size and then average it over all evaluation tasks to provide a single measure of stability. Moreover, Catastrophic Forgetting Resilience (CFR) is a key indicator of the stability-plasticity trade-off. Furthermore, the Forgetting Measure (FM) measures the negative influence that learning a task has on the test R@10 of all old tasks. It is the average forgetting of all old tasks, where forgetting of a task is the difference between its highest and current performance. Furthermore, BWT measures the positive influence of learning a new task on the test R@10 of old tasks. EBWT improves upon this metric by considering the average of the BWT computed after each task. Windowed-Plasticity 5 (WP5) measures the ability to learn new information by quantifying the maximum average test R@10 increase over a sliding window of 5 evaluations. Incremental Plasticity (IP) tracks changes in validation R@10 for each task over time.

However, it is important to note that our lifelong learning metrics have different interpretations in the Merge All dataset construction strategy, which differs from the traditional task-incremental setup. To our knowledge, an interpretation of these metrics in this setting has not been thoroughly conducted. As such, we propose that metrics should be interpreted with an understanding that they may reflect an adaptation to gradual shifts in data distribution rather than abrupt task changes. Specifically, WF5 may reflect not just forgetting old tasks but also the ability to balance and retain knowledge across an expanding dataset. WP5 could indicate how well the model adapts to the growing complexity of the combined dataset rather than purely learning new, isolated tasks. FM, in this context, may represent the ability to maintain performance on earlier data points as the dataset grows. EBWT might reflect the capacity to leverage newly added data to improve performance on the entire historical dataset. CFR becomes a measure of stability in the face of an expanding, potentially more complex dataset. IP may represent how quickly the model adapts to the evolving nature of the combined dataset rather than discrete new tasks. These metrics in the Merge All case measure the ability to accumulate and refine knowledge over time rather than strictly measuring performance on isolated tasks.

It is worth analyzing the effect of EWC. Our results in Sec. 4.3 suggest that the effect of EWC is not uniform across different task-ordering strategies. In curriculum-based ordering, EWC seems to improve plasticity (WP5 and IP) at the cost of stability and continuous improvement (WF5, FM, EBWT, and CFR). An exception is Setup 7, which improves WP5 and FM. This suggests that the Merge All strategy creates a more nuanced balance between stability and plasticity. EWC generally improves stability and plasticity metrics, except IP, in the popularity order for the Single Repository strategy. This may be because this ordering is less optimized for learning, and EWC helps to mitigate some of its shortcomings. However, when used with a more effective curriculum-based ordering, EWC interferes with the carefully structured learning process, leading to mixed results. Moreover, in the Merge All scenario, EWC offers benefits only for the curriculum learning setups, suggesting that its effectiveness might be limited in more complex, merged datasets. This can be explained by the fact that the Merge All strategy is a memory-based approach to lifelong learning. As such, using both EWC and the Merge All strategy may lead to excessive stability. This analysis further explains why LeanAgent's setup is superior.

## A.5    FURTHER *sorry* THEOREM PROVING DISCUSSION

We provide some additional discussion on the results in Table 2.

First, we note that comparing LeanAgent to ReProver+, the fine-tuning baseline, is not a direct comparison. Specifically, we note how mathematicians often formalize across multiple domains and projects simultaneously or cyclically. We use this as motivation for connecting mathematical knowledge between domains. Moreover, a key use case is formalizing new repositories without retraining on the entire dataset each time. When a mathematician creates some new repositories and adds them to a curriculum, they can simply progressively train LeanAgent on the new repositories while maintaining performance on previous ones, as shown in our experiments with both the initial

Table 10: Accuracy comparison across setups. Accuracy is calculated as (proven theorems / total *sorry* theorems). MIL denotes the Mathematics in Lean Source repository, LA denotes LeanAgent, and SX denotes Setup X (e.g., S1 is Setup 1). The best accuracy for each repository is in bold.

| Repository | #*sorry*s | Accuracy (%) | | | | | | | |
|---|---|---|---|---|---|---|---|---|---|
| | | **LA** | **S1** | **S2** | **S3** | **S4** | **S5** | **S6** | **S7** |
| MIL | 29 | **72.4** | 55.2 | 41.4 | 55.2 | 55.2 | 48.3 | 58.6 | 48.3 |
| SciLean | 294 | **9.2** | 8.5 | 7.5 | 6.8 | 8.5 | 8.5 | 8.8 | 7.8 |
| PFR | 37 | **2.7** | 0.0 | 0.0 | 0.0 | 0.0 | 0.0 | 0.0 | 0.0 |

curriculum and sub-curriculum. This is much more practical than retraining on the entire dataset, which would be expensive in terms of compute and time, especially as the number of repositories grows.

Moreover, data scarcity is a major problem in theorem proving. As such, having enough high-quality data for effective pre-training on all repositories may not be feasible. Training on all data, an approach similar to existing work, could prevent the model from generalizing to new repositories. However, LeanAgent does not have this restraint as it continuously generalizes to and improves on ever-expanding mathematical knowledge without forgetting previously learned knowledge.

In addition, the results in Table 4 show that our lifelong learning setup leads to effective backward transfer. This is a strong advantage that pre-training does not provide and is also a reason why LeanAgent demonstrates progressive learning, starting with basic concepts and advancing to more complex ones. Also, although not a direct comparison to the pre-training approach, the "Merge All" strategy indicates decreased performance over time. This dataset strategy can be interpreted as being closer to pre-training than the "Single Repository" strategy, suggesting lower than desired pre-training performance when training on all data.

Second, we note that LeanAgent improves over the direct fine-tuning baseline on various repositories, including MiniF2F, Scilean, and PFR. We find that these repositories contain a range of mathematical concepts that require progressively more advanced reasoning capabilities. This highlights the effectiveness of our lifelong learning approach in continuously generalizing to and improving on expanding mathematical knowledge without catastrophic forgetting.

In addition to comparing LeanAgent and ReProver, we conduct an ablation study between LeanAgent and the seven variants discussed in Sec. 4.3 regarding the original curriculum. The detailed *sorry* theorem proving comparison, which focuses on some of the repositories compared in Sec. 4.2, is in Table 10. Note that when using the Merge All strategy, only *sorry* theorems from the new repository are proven during each iteration of lifelong learning. We devote the rest of this section to the detailed comparison of *sorry* theorems that these setups can prove.

**Mathematics in Lean Source.** We notice a progression in LeanAgent's proving ability in the Mathematics in Lean Source repository. During lifelong learning, LeanAgent demonstrates a grasp of fundamental algebraic structures and basic mathematical operations:

a) Group and Ring Theory: LeanAgent proves theorems about basic algebraic structures. For instance, `MyGroup.mul_right_inv` shows that multiplying an element by its inverse yields the identity and `MyRing.add_right_cancel` demonstrates the cancellation property in ring addition.

```
theorem mul_right_inv (a : G) : a * a⁻¹ = 1 := by
    simp

theorem add_right_cancel {a b c : R} (h : a + b = c + b) : a = c := by
    simpa using h
```

b) Elementary Number Theory: LeanAgent handles fundamental arithmetic properties, including `MyRing.zero_mul`, which proves that zero multiplied by any number is zero, and `MyRing.neg_neg`, which shows that the negative of a negative number is the original number.

```
theorem zero_mul (a : R) : 0 * a = 0 := by
    rw [MulZeroClass.zero_mul]

theorem neg_neg (a : R) : - -a = a := by
    simp
```

c) Order Theory: LeanAgent grasps order theory, as evidenced by `absorb1`, which proves that the infimum of x and the supremum of x and y is always equal to x, and `absorb2`, which demonstrates that the supremum of x and the infimum of x and y is always equal to x.

```
theorem absorb1 : x ⊓ (x ⊔ y) = x := by
    simp

theorem absorb2 : x ⊔ x ⊓ y = x := by
    simp
```

d) Rudimentary Real Analysis: LeanAgent demonstrates an early capability to handle properties related to real numbers and absolute values, as shown by `C03S05.MyAbs.abs_add`, which proves the triangle inequality for real numbers.

```
theorem abs_add (x y : ℝ) : |x + y| ≤ |x| + |y| := by
    apply abs_add_le
```

10/14 of these proven *sorry* theorems from Mathematics in Lean Source during the lifelong learning process are from the exercise file for proving identities about algebraic structures. This indicates that LeanAgent starts its grasp of mathematical concepts from the basics.

Crucially, by the end of the lifelong learning process, LeanAgent exhibits significant growth in its mathematical reasoning abilities:

a) Quantifier Manipulation: LeanAgent exhibits more advanced logical reasoning by managing multiple quantifiers and implications, as evidenced by `C03S01.my_lemma3`, which proves a complex statement involving bounds and absolute values with multiple quantifiers and conditions, and `C03S05.MyAbs.abs_lt`, which establishes that the absolute value of x being less than y is equivalent to $-y < x \land x < y$.

```
theorem my_lemma3 :
        ∀ {x y ε : ℝ}, 0 < ε → ε ≤ 1 → |x| < ε → |y| < ε → |x * y| < ε := by
    apply C03S01.my_lemma

theorem abs_lt : |x| < y ↔ -y < x ∧ x < y := by
    cases x
    exact abs_lt
```

b) Set Theory and Relations: LeanAgent handles abstract set-theoretic concepts, as shown by `C03S01.Subset.trans`, which proves that subset relations are transitive.

```
theorem Subset.trans : r ⊆ s → s ⊆ t → r ⊆ t := by
    exact Set.Subset.trans
```

Now, only 2/7 *sorry* theorems from Mathematics in Lean Source are from the exercise file for proving identities about algebraic structures. This suggests that lifelong learning allowed LeanAgent to transition to gaining a stronger ability to work with premises for more complicated proofs.

We gain some insights from comparing the performance of LeanAgent over time on Mathematics in Lean Source to other setups. For example, the fact that ReProver can handle harder theorems out of the box, such as `C03S01.my_lemma3`, but fewer theorems overall suggests that it has a broader knowledge base initially but loses performance from a lack of adaptability. Furthermore, Setup 5 proves the same *sorry* theorems as LeanAgent does during lifelong learning. This suggests that pure curriculum learning without EWC or the Merge All strategy emphasizes grasping easier concepts earlier. This mimics the insights gained from the lifelong learning analysis. However, Setups 3 and 7 (curriculum with EWC and/or Merge All) demonstrate some knowledge plasticity, proving harder theorems during lifelong learning, such as `C03S01.Subset.trans` in Setup 3. However, this comes at the cost of proving basic theorems, showing catastrophic forgetting. For example, Setups 3 and 7 could not prove the trivial theorems `MyGroup.mul_right_inv` and `MyRing.zero_mul`, respectively, during lifelong learning, whereas LeanAgent could. This again aligns with the insights from the lifelong learning scores from our previous analysis.

By the end of lifelong learning, the *sorry* theorems that LeanAgent prove from Mathematics in Lean Source are a superset of those that the other setups prove. This shows that LeanAgent's lifelong learning setup provides continuously improving capabilities to reason about more advanced premises and proofs than other setups. For example, LeanAgent is the only system, except for Setup 6, which can prove theorem `C03S05.MyAbs.abs_lt`. LeanAgent achieved this by using the available premises, such as `abs_lt` with a statement similar to `C03S05.MyAbs.abs_lt`.

An interesting case study can be found in the dichotomy between the theorems `C03S05.MyAbs.neg_le_abs_self` and `C03S05.MyAbs.le_abs_self`. LeanAgent can prove `C03S05.MyAbs.neg_le_abs_self` by referencing `C03S05.MyAbs.le_abs_self`, which is still unproven at that point:

```
theorem neg_le_abs_self (x : ℝ) : -x ≤ |x| := by
    simpa using C03S05.MyAbs.le_abs_self (-x)
```

At the end of lifelong learning, LeanAgent can prove `C03S05.MyAbs.le_abs_self`:

```
theorem le_abs_self (x : ℝ) : x ≤ |x| := by
    rw [le_abs]
    simp
```

It achieves this through its use of the `le_abs` premise, which provides conditions for when an element is less than or equal to the absolute value of another. This suggests that LeanAgent begins by using existing knowledge where possible before trying to realizing why existing facts are reasonable.

**SciLean.** We examine the *sorry* theorems from SciLean that LeanAgent proved to gain some further key insights about its performance. During the lifelong learning process, LeanAgent demonstrated stronger understanding relative to ReProver in a wide range of mathematical concepts from SciLean. These theorems primarily focus on:

a) Fundamental Algebraic Structures: LeanAgent proves basic algebraic operations and properties, such as `SciLean.scalar_div_one`, which proves that dividing any number by one yields the same number, `SciLean.scalar_min_zero_one`, which demonstrates the minimum value between 0 and 1 is 0, and `Function.invFun.id_rule`, which proves that the inverse of the identity function is the identity function itself.

```
theorem scalar_div_one (x : R) : x / 1 = x := by
    simp

theorem scalar_min_zero_one  : min (0 : R) (1 : R) = 0 := by
    rw [min_comm]
    simp

theorem id_rule : invFun (fun (x : X) => x) = fun x => x := by
    apply Function.invFun_comp
    exact Function.injective_id
```

b) Linear and Affine Maps: LeanAgent handles basic properties of linear and affine maps effectively, recognizing their structure in `IsLinearMap.isLinearMap_apply`, which proves the linearity of function applications, and `IsAffineMap.IsAffineMap_apply`, which demonstrates the affine property of function applications.

```
theorem isLinearMap_apply (i : ι) : IsLinearMap R (fun f : (i : ι) → E i ↦ f i) := by
    constructor
    all_goals aesop

theorem IsAffineMap_apply (i : ι) : IsAffineMap R (fun f : (i : ι) → E i ↦ f i) := by
    constructor
    constructor
    simp
    simp
```

c) Measure Theory Basics: LeanAgent starts grasping measure theory concepts, exemplified by `SciLean.ite_pull_measureOf`, which handles conditional measure selection between two measures based on a proposition, `SciLean.Measure.prod_volume`,

which proves that the product of two volume measures is the volume measure itself, and `SciLean.ite_pull_ennreal_toReal`, which proves that conditionally pulling out an extended non-negative real and converting it to a real is equivalent to converting the individual components first.

```
theorem ite_pull_measureOf {X} [MeasurableSpace X] (c : Prop) [Decidable c]
      (μ ν : Measure X) (A : Set X) :
      (if c then μ else ν) A
      =
      (if c then μ A else ν A) := by
  split_ifs <;> rfl

theorem Measure.prod_volume {X Y} [MeasureSpace X] [MeasureSpace Y]  :
      (Measure.prod (volume : Measure X) (volume : Measure Y)) = volume := by
  rfl

theorem ite_pull_ennreal_toReal (c : Prop) [Decidable c] (x y : ENNReal)  :
      (if c then x else y).toReal
      =
      (if c then x.toReal else y.toReal) := by
  split_ifs <;> rfl
```

d) Floating-Point Operations: LeanAgent demonstrates an early grasp of floating-point representations and their correspondence to real numbers, shown by `SciLean.re_float`, proving that a floating-point number's real-like part is itself.

```
theorem re_float (a : Float) : RCLike.re a = a := by
    exact RCLike.re_eq_self_of_le le_rfl
```

The proofs during this phase are characteristically concise, often using basic tactics like `simp`, `rfl`, or `aesop` that do not use premises. This suggests that LeanAgent recognizes these theorems are straightforward enough to prove without the complex retrieval of premises.

Crucially, by the end of the lifelong learning process, LeanAgent exhibits significant growth in its mathematical reasoning abilities on SciLean, just as it did with Mathematics in Lean Source:

a) Advanced Function Spaces: LeanAgent understands concepts in advanced function spaces, such as `SciLean.ContCDiffMapFD_eta`, which demonstrates the eta reduction property for continuously differentiable maps over finite dimensions.

```
theorem ContCDiffMapFD_eta (f : X →FD[K,n] Y) : (fun x →FD[K,n] f x) = f := by
    simp only [DFunLike.ext_iff]
    aesop
```

b) Sophisticated Bijections: LeanAgent grows in its ability to work with product spaces and bijections, proving theorems such as `Function.Bijective.Prod.mk.arg_fstsnd.Bijective_rule_simple'`, which proves the bijectivity of a function that swaps elements in a product space and `Function.Bijective.Equiv.invFun.arg_a0.Bijective_rule`, which proves that the composition of a bijection and its inverse remains bijective. Crucially, these theorems might seem simple, but they demonstrate LeanAgent's capability to handle abstract algebraic thinking.

```
theorem Prod.mk.arg_fstsnd.Bijective_rule_simple'
    : Bijective (fun xy : X×Y => (xy.2, xy.1))
    := by
  constructor <;> intro h
  all_goals aesop

theorem Equiv.invFun.arg_a0.Bijective_rule (f : Y ≃ Z) (g : X → Z) (hf : Bijective g)
    : Bijective (fun x => f.invFun (g x)) := by
  convert hf
  simp [hf]
  exact f.symm.bijective.comp hf
```

c) Abstract Algebraic Structures: LeanAgent proves further abstract algebraic properties, including `SciLean.CDifferentiable.id_rule`, which proves that the identity function is continuously differentiable.

```
theorem CDifferentiable.id_rule : CDifferentiable K (fun x : X => x) := by
    intro x
    unfold SciLean.CDifferentiableAt
    tauto
```

d) Data Structures in Mathematics: LeanAgent navigates proofs involving array types in a mathematical context, proving theorems such as `SciLean.ArrayType.ext`, which proves that two arrays are equal if their elements are equal at all indices.

```
theorem ext (x y : Cont) : (∀ i, x[i] = y[i]) → x = y := by
    intro h
    apply SciLean.ArrayType.get_injective
    simp only [h]
```

LeanAgent proved this theorem about a traditionally computer-science-oriented data structure from a mathematical lens, which some other setups could not do.

The proofs at this stage are more sophisticated, involving multiple steps and combining various mathematical concepts. This indicates a deeper ability to connect different areas of mathematics.

The progression from basic algebraic structures to advanced function spaces and data structures (like array types) shows that LeanAgent is bridging the gap between pure mathematical concepts and their applications in computational mathematics. Furthermore, the progression from basic integral manipulations to advanced function spaces indicates that LeanAgent is improving its premise selection over time. It learns to identify and apply more sophisticated mathematical structures and theorems as premises.

By the end of lifelong learning, the *sorry* theorems that LeanAgent proves from SciLean are almost entirely a superset of those that the other setups prove. This corroborates our previous assertion from our analysis of the *sorry* theorems from Mathematics in Lean Source that LeanAgent's lifelong learning setup provides it with continuously improving capabilities to reason about premises and proofs that outperform other setups.

Crucially, LeanAgent could prove `re_float` during lifelong learning while no other setup could. This indicates that LeanAgent's more measured and stable approach allowed it to grasp floating-point representations and their relation to reals. At the same time, other setups prioritized this less with their reduced stability. This may also suggest that continuous improvement allowed LeanAgent to process new and unique concepts from new repositories.

We gain some interesting insights from comparing the performance of LeanAgent over time on SciLean to other setups. For example, the fact that ReProver could not prove the trivial theorem `SciLean.ite_pull_ennreal_toReal` while LeanAgent could, even during lifelong learning, suggests that this baseline cannot handle foundational concepts. LeanAgent has a better grasp of these concepts from lifelong learning. This is due to the increased stability of LeanAgent and improvement from learning a new task, as shown in the lifelong learning metric analysis in Sec. 4.3.

Furthermore, an intriguing observation is that Setup 3 could prove `Function.Bijective.Prod.mk.arg_fstsnd.Bijective_rule_simple'` during lifelong learning. However, as mentioned above, LeanAgent could only prove this theorem at the end of lifelong learning. This suggests that Setup 3 is more plastic during lifelong learning, while LeanAgent remains more stable. This corroborates the analysis from the lifelong learning metrics. However, this again comes at the cost of grasping basic theorems. For example, Setup 3 cannot prove the trivial measure theory theorem `SciLean.Measure.prod_volume`, whereas LeanAgent could during lifelong learning, again suggesting that it favors plasticity over stability.

An interesting case is that LeanAgent can prove `SciLean.norm₂_scalar` during lifelong learning but not `SciLean.norm2_scalar`. Conversely, Setup 2 proved `SciLean.norm2_scalar` but not `SciLean.norm₂_scalar`.

```
theorem norm₂_scalar {R} [RealScalar R] (x : R) :
        ‖x‖₂[R] = Scalar.abs x := by
    rw [SciLean.scalar_norm]

theorem norm2_scalar {R} [RealScalar R] (x : R) :
        ‖x‖₂²[R] = x^2 := by
    symm
    simp [sq]
    congr
    simp
```

Setup 2 uses the `sq` premise from `mathlib4`, which states that the square of an element is the same as multiplying that element by itself, while LeanAgent used the `SciLean.scalar_norm` premise from SciLean, which states that the 2-norm of a real scalar is equal to the absolute value of that scalar. This suggests that LeanAgent prefers to use new premises if possible rather than simplistic pretrained ones. This also makes sense since Setup 2 uses popularity order, which generally provides poor stability and plasticity.

**PFR.** Crucially, LeanAgent is the only setup to prove a *sorry* theorem from PFR. We can prove `condRho_of_translate`, which states a form of a translation invariance property of randomness measures.

```
lemma condRho_of_translate {Ω S : Type*} [MeasureSpace Ω] (X : Ω → G) (Y : Ω → S)
        (A : Finset G) (s:G) : condRho (fun ω ↦ X ω + s) Y A = condRho X Y A := by
    simp only [condRho, rho_of_translate]
```

LeanAgent suggests that the proof is straightforward after expanding definitions of condRho and considering how rho, the randomness measure, behaves under translation (as captured by the `rho_of_translate` lemma). The fact that LeanAgent could trivially identify such a short proof using existing premises while the maintainers of the PFR repository did not suggests the power of our approach. This suggests that by learning the foundations of the PFR lemmas using its improved stability, LeanAgent was able to grasp some basic definitions.

However, LeanAgent and other setups could not prove any PFR theorems after lifelong learning. This suggests that LeanAgent requires more training time or data to further strengthen its knowledge in this new area of mathematics.

**Alternate PFR Commit.** We also analyze whether LeanAgent can generalize to a different commit of a repository in the curriculum. We choose commit 861715b9bf9482d2442760169cb2a3ff54091f75, because PFR/RhoFunctional.lean, the file from which we proved `condRho_of_translate` in the newer commit, did not exist in the old commit. This allowed the *sorry* theorems in the old commit to be more distinct. It proved two theorems, including the theorem `multiDist_copy` about the equality of distributions when copying random variables across different measure spaces that ReProver could not. LeanAgent achieved this with just the tactic `rfl`. However, these statements were about `multiDist`, another

*sorry* theorem. Since any two instances of *sorry* are automatically equal by `rfl`, LeanAgent exploits this technicality to prove the two theorems.

```
lemma multiDist_copy {m:ℕ} {Ω : Fin m → Type*} {Ω' : Fin m → Type*}
        (hΩ : (i : Fin m) → MeasureSpace (Ω i))
        (hΩ': (i : Fin m) → MeasureSpace (Ω' i)) (X : (i : Fin m) → (Ω i) → G)
        (X' : (i : Fin m) → (Ω' i) → G)
        (hident: ∀ i, IdentDistrib (X i) (X' i) (hΩ i).volume (hΩ' i).volume) :
        D[X ; hΩ] = D[X' ; hΩ'] := by
    rfl

lemma multiDist_of_perm {m:ℕ} {Ω: Fin m → Type*}
        (hΩ : (i : Fin m) → MeasureSpace (Ω i))
        (X : (i : Fin m) → (Ω i) → G) (φ : Equiv.Perm (Fin m)) :
        D[X ; hΩ] = D[fun i ↦ X (φ i); fun i ↦ hΩ (φ i)]:= by
    rfl
```

This commit also provides another interesting example. The PFR maintainers used `0 = 1` as a placeholder for some *sorry* theorems, five of which are `multiTau_min_exists`, `multiTau_min_sum_le`, `sub_multiDistance_le`, `sub_condMultiDistance_le`, `sub_condMultiDistance_le'`. Interestingly, LeanAgent finds unintended constructs and proves these theorems with this proof:

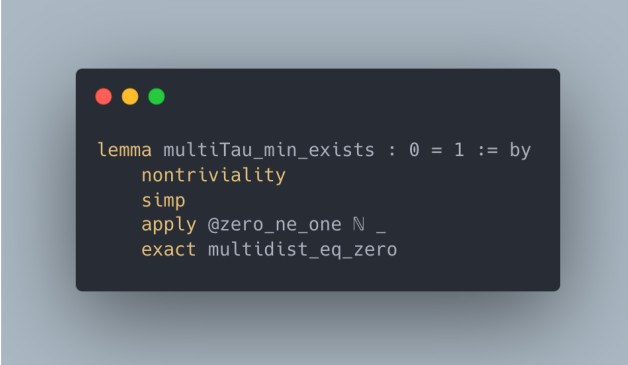

It combines the known fact that $0 \neq 1$ with the placeholder theorem `multidist_eq_zero`, which states $0 = 1$. As such, it proves `False`, allowing it to derive anything, including $0 = 1$.

**MiniF2F.** This repository consists of a validation and test split. Prior work evaluated performance as the number of *sorry* theorems proved and a Pass@k metric on the test split (Yang et al., 2023). However, given that LeanAgent is a framework, not a model, quantitatively comparing it with existing methods using a Pass@k metric is misleading. In line with our existing setups, we do not treat MiniF2F as a benchmark. Instead, we disregard its existing splits and compare LeanAgent's proven *sorry* theorems with those of ReProver.

LeanAgent can prove 99 theorems, while ReProver can only prove 85. As mentioned in Sec. 4.2, we append MiniF2F to the initial curriculum to demonstrate the use case of formalizing a new repository in parallel with the ones in the starting curriculum. As such, interesting observations can be made by comparing ReProver (starting point) with LeanAgent (ending point). The types of *sorry* theorems LeanAgent can prove demonstrate its increasing understanding relative to ReProver in complex mathematical concepts due to lifelong learning.

ReProver initially demonstrated proficiency in a range of foundational mathematical areas on MiniF2F:

a) Basic Arithmetic and Number Theory: ReProver could handle simple arithmetic and modular arithmetic problems, such as `mathd_numbertheory_254`, a theorem about modular arithmetic and basic addition, `mathd_numbertheory_342`, a theorem about basic divisibility, and `mathd_algebra_304`, a theorem about simple exponentiation. These proofs only rely on the `norm_num` tactic, which evaluates arithmetic expressions. This suggests a less sophisticated proving ability of mathematics at the start of lifelong learning.

```
theorem mathd_numbertheory_254 : (239 + 174 + 83) % 10 = 6 := by
    norm_num

theorem mathd_numbertheory_342 : 54 % 6 = 0 := by
    norm_num

theorem mathd_algebra_304 : 91^2 = 8281 := by
    norm_num
```

b) Elementary Algebra: ReProver could solve basic algebraic equations and perform straightforward manipulations, such as `mathd_algebra_141`, which proves a statement about quadratic expressions, `mathd_algebra_329`, which shows a grasp of systems of linear equations, and `mathd_algebra_547`, which proves basic algebraic manipulation with roots.

```
theorem mathd_algebra_141
    (a b : ℝ)
    (h₁ : (a * b)=180)
    (h₂ : 2 * (a + b)=54) :
    (a^2 + b^2) = 369 := by
    nlinarith

theorem mathd_algebra_329
    (x y : ℝ)
    (h₀ : 3 * y = x)
    (h₁ : 2 * x + 5 * y = 11) :
    x + y = 4 := by
    linarith

theorem mathd_algebra_547
    (x y : ℝ)
    (h₀ : x = 5)
    (h₁ : y = 2) :
    Real.sqrt (x ^ 3 - 2 ^ y) = 11 := by
    simp [h₀, h₁, sq]
    rw [Real.sqrt_eq_iff_sq_eq] <;> norm_num
```

c) Basic Calculus and Analysis: ReProver showed early capabilities in dealing with logarithms and exponentials, including `mathd_algebra_484`, a theorem involving dividing logarithmic expressions.

```
theorem mathd_algebra_484 : Real.log 27 / Real.log 3 = 3 := by
    field_simp
    rw [← Real.log_rpow]
    all_goals norm_num
```

Notably, the proofs at this stage were characteristically concise, often using basic tactics like `norm_num`, `linarith`, and `field_simp`. This suggests that ReProver recognized these theorems as straightforward enough to prove without complex retrieval of premises, similar to its behavior with previous repositories.

However, by the end of the lifelong learning process, LeanAgent exhibited significant growth in its mathematical reasoning abilities on MiniF2F:

a) Advanced Number Theory: LeanAgent showed a more advanced grasp of number theory, proving theorems like `mathd_numbertheory_293`, a complex theorem about divisibility involving a complex expression and `mathd_numbertheory_233`, a theorem dealing with modular arithmetic in $\text{ZMod}(11^2)$.

```
theorem mathd_numbertheory_293
    (n : ℕ)
    (h₀ : n ≤ 9)
    (h₁ : 11|20 * 100 + 10 * n + 7) :
    n = 5 := by
    omega

theorem mathd_numbertheory_233
    (b :  ZMod (11^2))
    (h₀ : b = 24⁻¹) :
    b = 116 := by
    exact h₀
```

b) Sophisticated Algebra: LeanAgent showed a better grasp of more complex algebraic manipulations. Theorems include mathd_algebra_148, which involves function definitions and solving for unknown coefficients, and amc12a_2016_p3, involving a special case of a function.

```
theorem mathd_algebra_148
    (c : ℝ)
    (f : ℝ → ℝ)
    (h₀ : ∀ x, f x = c * x^3 - 9 * x + 3)
    (h₁ : f 2 = 9) :
    c = 3 := by
    linarith [h₀ 2]

theorem amc12a_2016_p3 (f : ℝ → ℝ → ℝ)
    (h₀ : ∀ x, ∀ (y) (_ : y ≠ 0), f x y = x - y * Int.floor (x / y)) :
    f (3 / 8) (-(2 / 5)) = -(1 / 40) := by
    norm_num [h₀]
    field_simp
    norm_cast
```

c) Advanced Calculus and Analysis: LeanAgent demonstrated improved capabilities in handling more complex analytical problems, including mathd_algebra_270, a theorem involving function composition and rational expressions.

```
theorem mathd_algebra_270
    (f : ℝ → ℝ)
    (h₀ : ∀ x, x ≠ -2 -> f x = 1 / (x + 2)) :
    f (f 1) = 3/7 := by
    set_option tactic.skipAssignedInstances false in norm_num [h₀]
```

d) Complex Induction: LeanAgent became adept at more advanced induction proofs. An example is `induction_12dvd4expnp1p20`, a theorem about divisibility that requires an induction proof.

```
theorem induction_12dvd4expnp1p20 (n : ℕ) : 12 | 4^(n+1) + 20 := by
    norm_num
    induction' n with n hn
    simp
    omega
```

e) Complex Quantifiers and Inequalities: LeanAgent increased its ability to prove more complex logical statements, such as `amc12a_2002_p6`, a theorem involving multiple existential quantifiers and inequalities.

```
theorem amc12a_2002_p6 (n : ℕ) (h₀ : 0 < n) : ∃ m, (m > n ∧ ∃ p, m * p ≤ m + p) := by
    lift n to ℕ+ using h₀
    cases' n with n
    exact ⟨_, lt_add_of_pos_right _ zero_lt_one, 1, by simp⟩
```

The proofs at this later stage are more sophisticated, usually involving multiple steps and combining various mathematical concepts or indicating a better ability to connect different areas of mathematics, mirroring the progression observed in Mathematics in Lean Source and SciLean. For example, as shown above, LeanAgent provides a one-line proof to the relatively advanced theorem `mathd_numbertheory_233`. The proof means the hypothesis directly proves the goal. This suggests that LeanAgent grasps modular arithmetic and recognizes when a given hypothesis is sufficient to prove the goal without additional steps.

Furthermore, as shown above, LeanAgent uses four tactics to prove the `induction_12dvd4expnp1p20` theorem. This demonstrates its ability to handle more complex number theory proofs and use advanced tactics. This again shows that LeanAgent can recognize when it does not require complex premise retrieval.

LeanAgent demonstrates a similar grasp of the theorem `amc12a_2002_p6`. Notably, it combines the simple premises `lt_add_of_pos_right`, which describes how an element is less than that element added with a positive one, and `zero_lt_one`, which states that 0 is less than 1, with more advanced tactics like `lift`, `cases`, and `exact` with complex term construction. This demonstrates its ability to reuse foundational concepts for more complex proofs of abstract mathematical concepts, showing its stability.

Importantly, LeanAgent's performance on MiniF2F showcases its ability to adapt and improve across different mathematical domains. We see this in the progression from ReProver's basic arithmetic and algebra to LeanAgent's more advanced number theory, calculus, and abstract algebra. This aligns with the observations from Mathematics in Lean Source and SciLean, further supporting the effectiveness of LeanAgent's lifelong learning approach in theorem proving across various mathematical repositories.

Furthermore, early proofs from ReProver dealt with concrete numbers and simple equations. Later proofs from LeanAgent involved more abstract concepts like equivalence relations and function properties. LeanAgent gained the capability to handle more complex number theory problems involving divisibility under constraints. Moreover, LeanAgent shifted from solving basic linear and quadratic equations to analyzing functions and their compositions. Also, early proofs often used `norm_num` for straightforward computations. Later proofs employed more varied tactics and premises, suggesting a more sophisticated approach to proof construction. This all crucially suggests that while existing methods, like ReProver, may be more tailored to simpler computation problems, LeanAgent is superior on complex and analytical problems. These are precisely the types of problems present in advanced mathematics. This also corroborates LeanAgent's performance on the repositories mentioned previously.

In addition, we evaluate LeanAgent on the Lean4 version of the MiniF2F test set using the same experimental setup from prior experiments, taking care not to progressively train on Test.lean and only proving *sorry* theorems from it. The results are as follows:

LeanAgent: Pass@1 of 38.1% (93 / 244 theorems)

ReProver (our run on Lean4): Pass@1 of 34.0% (83 / 244 theorems)

ReProver was only tested on the Lean3 test set in the LeanDojo paper, so we ran ReProver on the Lean4 test set for a fairer comparison. TheoremLlama (Wang et al., 2024c) reports a 33.61% cumulative accuracy on the Lean4 test set, but this makes a direct comparison with the Pass@1 rates of LeanAgent and ReProver infeasible. Moreover, DeepSeek-Prover-V1.5 (Xin et al., 2024b) reports a state-of-the-art result of 63.5% on the Lean4 test set.

However, LeanAgent is a lifelong learning framework, not a model. The performance of LeanAgent is dependent on the retriever used as the starting point - ReProver's retriever in our case. As such, the metrics for other methods besides ReProver cannot be directly compared with LeanAgent. Such a comparison would be impractical for reasons including differences in data, pretraining, and fine-tuning. Again, we only compare with ReProver because we use ReProver's retriever as the starting one in LeanAgent, allowing for a more faithful comparison. We use ReProver because past work has shown that retrieval leads to improved generalizability.

Moreover, we design LeanAgent to continuously generalize to and improve on ever-expanding mathematical knowledge without forgetting previously learned knowledge. Our goal is not to beat the MiniF2F benchmark; instead, we aim to perform well in proving *sorry* theorems across a range of diverse repositories. The other approaches mentioned focus on different objectives and don't address the lifelong learning aspect that is central to our work.

**Formal Book.** We first examine the *sorry* theorems from Formal Book that LeanAgent proved during lifelong learning. These theorems centered around:

a) Real Analysis and Inequalities: LeanAgent demonstrates a better understanding relative to ReProver in real number properties and can handle basic inequality reasoning, proving `book.irrational.lem_aux_ii`, which involves real analysis and inequalities.

```
lemma lem_aux_ii (n : ℕ) (x : ℝ) (h_1 : 0 < x) (h_2 : x < 0) :
        (0 < f_aux n x) ∧ (f_aux n x < (1 : ℝ) / n.factorial) := by
    constructor <;> linarith
```

b) Number Theory: LeanAgent shows capabilities in fundamental number theory concepts, proving `book.quadratic_reciprocity.quadratic_reciprocity_2`, a key result in quadratic reciprocity.

```
theorem quadratic_reciprocity_2 (p q : ℕ) (hp : p ≠ 2) (hq : q ≠ 2)
    [Fact (Nat.Prime p)] [Fact (Nat.Prime q)] :
    (legendre_sym p q) * (legendre_sym q p) = -1 ^ ((p-1) / 2 * (q - 1) / 2 ) := by
    exact book.quadratic_reciprocity.quadratic_reciprocity_1 p q hp hq
```

Notably, the proof of the first theorem uses no premises, and the proof of the second uses a simple statement of quadratic reciprocity. However, by the end of the lifelong learning process, LeanAgent exhibits growth in its proving abilities in this repository:

a) Advanced Abstract Algebra: LeanAgent shows advancement in proving a key result in abstract algebra, `wedderburn` (Wedderburn's Little Theorem), which is a crucial result in abstract algebra, stating that every finite division ring is a field.

```
theorem wedderburn (h: Fintype R): IsField R := by
    apply Field.toIsField
```

LeanAgent's proof of the `wedderburn` theorem represents the ability to handle algebraic structures. By using the `Field.toIsField` premise, LeanAgent shows that it has grasped how to use the knowledge of what makes a ring a field. This requires an understanding of ring theory and field properties.

**Coxeter.** LeanAgent could not prove *sorry* theorems from the Coxeter repository during lifelong learning. However, by the end of lifelong learning, LeanAgent demonstrates a growing understanding of more complex algebraic structures, proving the lemma `invmap.of_eq` about Coxeter systems, again showing the ability to work with advanced concepts in group theory and abstract algebra. This corroborates the handling of abstract algebra necessary to prove the `wedderburn` theorem.

```
lemma invmap.of_eq {S:Set G} [CoxeterSystem G S] {s :S} : invmap S s = s := by
    simp [CoxeterSystem.Presentation.invmap]
    unfold CoxeterSystem.toMatrix
    apply CoxeterSystem.monoidLift.mapLift.of
```

LeanAgent's proof of `invmap.of_eq` involves unfolding definitions and applying specific properties of Coxeter systems. This demonstrates LeanAgent's growing understanding relative to ReProver in abstract algebra specific to the new repository it has learned from.

**Hairy Ball Theorem.** Moreover, LeanAgent could not prove *sorry* theorems from the Hairy Ball Theorem repository during lifelong learning. However, at the end of lifelong learning, LeanAgent again demonstrates a stronger understanding relative to ReProver in algebraic topology. It proves `HairyBallDiff`, which states a key step in the Hairy Ball Theorem, demonstrating a grasp of vector spaces, norms, and their connections to topological concepts.

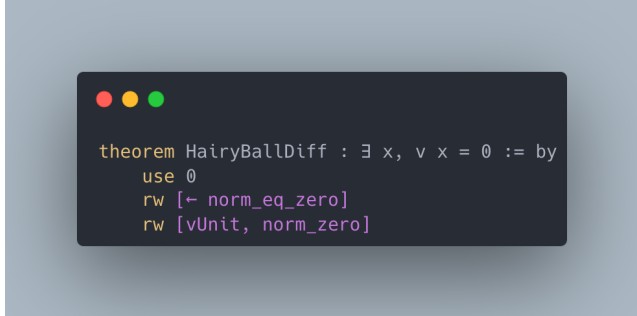

```
theorem HairyBallDiff : ∃ x, v x = 0 := by
    use 0
    rw [← norm_eq_zero]
    rw [vUnit, norm_zero]
```

Crucially, only LeanAgent could prove `invmap.of_eq`, `wedderburn`, and `HairyBallDiff`, demonstrating that it has developed much more advanced theorem-proving capabilities than other setups. These proofs show that LeanAgent can work with highly abstract concepts and apply them to specific mathematical objects.

### A.6 Curriculum Learning Analysis

In this section, we aim to answer the following questions: (1) Why does LeanAgent use $e^S$ ($S$ = number of proof steps) as its complexity metric? (2) Why does curriculum learning work in theorem proving? (3) Why does LeanAgent use curriculum learning instead of other lifelong learning methods?

**Complexity Measure.** There is no universal measure of proof complexity in Lean or other formal systems. One approach, the length-based measure, involves examining the proof length (number of steps or lines) and the size of the proof term in a formal system. While these can indicate verification complexity, they may not fully capture the complexity of discovering a proof (Arana & Stafford, 2023). Moreover, within the NLP literature, many works have related input length to complexity (Zaremba & Sutskever, 2015; Cirik et al., 2016; Spitkovsky et al.; Subramanian et al., 2017; Chang et al., 2021). Starting with shorter sequences and gradually increasing length improves model quality (Li et al., 2024).

These works demonstrate the gains from basing the complexity measure on input length. As such, we consider the equivalent of length in theorem proving to be the number of proof steps. However, we consider a linear scaling of length naive for theorem proving; it doesn't consider the combinatorial explosion of possible proof paths as the length of the proof increases. As such, we choose an

exponential scaling. Notably, a key strength of this choice is it is easy to compute and requires no additional hyperparameters to tune.

We now discuss some alternative complexity metrics and why we chose not to use them in Lean-Agent. One option is $e^B$, where $B$ represents the number of different proof paths that could be explored at each step. Formally, $B$ is defined as the average number of child nodes for each non-leaf node in the proof tree. This is sometimes called the branching factor. We refrain from using this complexity metric as computing this becomes computationally expensive for complex proofs. Moreover, another option is to consider the complexity of the theorem statement to determine complexity. For example, this could be measured by the number of unique symbols, the depth of nested expressions, or the number of quantifiers. However, developing a reliable metric for statement complexity that works across various mathematical domains could be challenging. Moreover, LeanAgent focuses on improving proof generation, so using a metric directly related to the proof process (number of steps) aligns better with this goal than statement complexity. Dependency-based complexity, where we order theorems based on their dependency structure within the mathematical library, wasn't used for multiple reasons. Namely, a theorem might depend on many simple results but still be relatively easy to prove, or it might depend on a few results but be very challenging. Furthermore, a topic-based curriculum would be unsuitable because LeanAgent aims to be a general-purpose framework. A topic-based approach might bias it towards certain mathematical domains. Moreover, a topic-based approach does not account for theorems spanning multiple mathematical concepts. Another option is a form of premise-based complexity measure. However, analyzing premise occurrence frequencies across repositories could be computationally expensive, especially for large repositories like SciLean.

**Curriculum Learning.** Prior work suggests that curriculum learning guides the learner towards better local minima in non-convex optimization problems (Bengio et al., 2009). Crucially, theorem proving involves navigating a highly non-convex optimization landscape, especially for complex mathematical statements. The space of possible proofs is vast and complex, with many potential dead ends and suboptimal solutions to parts of a proof. This makes theorem proving an ideal candidate for benefitting from curriculum learning. Moreover, curriculum learning has been shown to have an effect similar to unsupervised pre-training, acting as a form of regularization (Bengio et al., 2009). For theorem proving, curriculum learning allows LeanAgent to build a foundational knowledge base of mathematics before attempting more complex theorems, naturally leading to continuous improvement. This avoids suboptimal proof strategies early in training. Furthermore, this leads to more robust and generalizable proof techniques that work across a broader range of theorems, explaining LeanAgent's *sorry* theorem proving performance. Moreover, the ability to act as a regularizer means curriculum learning prevents the model from overfitting to specific types of proofs or mathematical domains. This allows for continuous generalizability, explaining LeanAgent's superior lifelong learning metric scores.

Moreover, other works from the literature show that curriculum learning biases models towards building constructive internal representations (Cirik et al., 2016). Specifically, it allows the model to use the knowledge from earlier steps in later predictions. In theorem proving, this allows Lean-Agent to learn basic proof skills, which then become building blocks for more complex proofs later on. This corroborates our analysis of LeanAgent's progression of proof complexity. Moreover, multiple past works agree that curriculum learning provides larger gains when training data is limited (Zaremba & Sutskever, 2015; Cirik et al., 2016; Spitkovsky et al.). This is the case in formal theorem proving, where the number of formalized theorems and proofs is limited. These past works also state that curriculum learning leads to better generalization, supporting the observations of LeanAgent's superior lifelong learning metrics.

This context supports LeanAgent's *sorry* theorem proving performance and superiority on lifelong learning metrics.

**Comparison with Other Lifelong Learning Methods.** Many other lifelong learning methods exist, such as those mentioned in Sec. 2. However, we chose curriculum learning for the following reasons. Regularization methods, like EWC, slow down learning on important parameters. However, the importance of parameters can change as the theorem complexity increases. This helps explain the lower *sorry* theorem proving performance of EWC methods and their performance on lifelong learning metrics. Moreover, memory-based techniques store examples from previous tasks to prevent forgetting. However, this can greatly affect these methods' balance of stability and plas-

ticity. This can be seen in the lifelong learning metrics of Merge All setups, including the negative IP values. Knowledge distillation requires a separate teacher model, but curriculum learning is more efficient as it provides the path to knowledge accumulation in a single model. Since LeanAgent is a framework, not a model, we refrain from using dynamic architecture adjustment to keep LeanAgent general to many LLM architectures. Moreover, recent work selectively updates parameters with the largest momentum magnitudes and uses selective reinitialization to maintain plasticity. However, these methods often focus on balancing performance across distinct tasks. In theorem proving, where tasks form a spectrum of increasing complexity, curriculum learning provides a more structured approach to knowledge accumulation.

## A.7 THEOREM PROVING PERFORMANCE SCORE (TPPS) ANALYSIS

In this work, we directly compared the number of proven theorems with ReProver. However, the field of theorem proving, especially in a lifelong learning context, currently lacks standardized performance metrics. To address this concern, this section discusses a possible alternative metric while acknowledging its limitations.

Theorem proving difficulty is inherently non-linear in nature. For example, LeanAgent's significantly improved performance over the baseline across multiple repositories allows it to prove progressively harder theorems. Furthermore, *sorry* theorems lack ground truth proofs, so proving one is valuable. To address these nuances, one could propose the Theorem Proving Performance Score (TPPS) to emphasize newly proven *sorry* theorems. Specifically, it could be stated that LeanAgent TPPS $=$ (# ReProver Theorems Proved) $+$ (# New Theorems Proved $* X$) $+ 1$, where $X$ represents the importance of proving a new theorem, and ReProver TPPS $=$ (# ReProver Theorems Proved) $+ 1$. Then, Improvement Factor $=$ (LeanAgent TPPS)/(ReProver TPPS).

The core idea behind TPPS is to assign a higher reward for newly proven theorems, aligning with lifelong learning objectives. However, it is important to recognize its preliminary nature and potential shortcomings. First, choosing the parameter $X$ is challenging. One approach is to choose a static value, such as $X = 10$, to standardize comparisons between LeanAgent and ReProver across diverse repositories. However, this may lead to inflated or deflated metrics on such repositories. Alternatively, $X$ could be chosen adaptively based on the difficulty of a repository, but this may similarly result in unrealistic metric scores. Overall, the TPPS metric focuses on quantity and faces challenges in ensuring comparisons across diverse repositories.

Moreover, the TPPS metric can be susceptible to artifacts that artificially inflate performance. For instance, several theorems in the PFR repository were proven on a technicality due to placeholder statements of "$0 = 1$", which could then be used to prove other "sorry" theorems through the principle of *ex falso quodlibet*. Examples such as this are due to the weakness of the current state of repositories rather than a fundamental shortcoming of our ML approach, underscoring the need for more robust and well-tested benchmarks.

Furthermore, TPPS does not account for the difficulty of the theorems being proved. While we initially considered using proof length as a proxy for difficulty (e.g., $e^S$ where $S$ is the number of proof steps), we found this approach problematic during proving evaluation. LeanAgent sometimes generates shorter proofs for difficult theorems, making proof length an unreliable indicator of theorem difficulty.

These issues expose broader challenges in evaluating theorem-proving systems. The field lacks standardized benchmarks, necessitating carefully curated sets of theorems with varying difficulties across different mathematical domains. Developing reliable metrics to assess theorem difficulty beyond simple measures like proof length or statement complexity is crucial. Future benchmarks should prevent the exploitation of technicalities. Moreover, metrics should consider the mathematical significance of proven theorems, not just their quantity. Finally, ensuring that performance metrics are meaningful and comparable across different mathematical repositories is essential for consistent evaluation.

In light of these considerations, we acknowledge that TPPS, while a step towards quantifying theorem-proving performance in a lifelong learning setting, has many limitations. Future work should focus on developing more sophisticated and robust evaluation frameworks that address these

challenges. For example, we plan to investigate more sophisticated measures that reward not only newly proved theorems but also difficult ones. By highlighting these issues, we hope to contribute to the ongoing discussion on how best to evaluate and compare theorem-proving systems, particularly in the context of lifelong learning. The development of standardized, reliable metrics will be crucial for measuring progress and guiding future research in this field.

