# OpenReview forum: "LeanAgent: Lifelong Learning for Formal Theorem Proving"
_ICLR.cc/2025/Conference — ICLR 2025 Poster_

### Official Review · Reviewer_W5az · 2024-10-31

**Soundness:** 2
**Presentation:** 2
**Contribution:** 2
**Rating:** 6
**Confidence:** 5

**Summary:**

This paper studies the topic of automated theorem proving in Lean language. It primarily builds on the method proposed by LeanDojo and considers the problem that a trained LLM may consider catastrophic forgetting when it is fine tuned on a new dataset. To address this issue, it suggests that LLMs should be trained for lifelong learning, where they learn new material after being fine-tuned without forgetting what they had learned before. It then considers several Lean repositories and tries to prove the unsolved theorems in those repositories. It reports vaguely some successes in solving some of the theorems.

**Strengths:**

The topic is very interesting and the experiments are on a diverse set of problems and repositories.

Literature review considers both the ML literature and math literature.

**Weaknesses:**

I found the presentation of results very disorganized and rather unusual in the context of the ML literature. Overall, the presentation seems to be a stream of consciousness.

I try to view the contribution from different perspectives, and I explain the shortcomings from each perspective.

1. The task of automated theorem proving:

    1.a. I tried for some time to find out what is the achieved accuracy of the proposed system on the miniF2F validation and test sets. I was shocked to see in the main body of the paper, in Tables 1 and 2, the paper provides the number of theorems in the datasets that it has proved without giving the total number of theorems nor the accuracy. I was expecting these tables to provide the accuracy of the proposed model on each of the datasets. I was only able to find somewhere deep in the appendix that the paper has been able to prove 99 theorems from the miniF2F dataset (not clear whether it is for the test set or valid set) which translates to accuracy of about 40%. It is as if the paper is trying to hide the accuracy on the known benchmark miniF2F.

    1.b. The accuracy of 40% on miniF2F does not appear to be an improvement in the recent literature. For example, the accuracy of Hypertree Proof Search from 2022 is 41% on the miniF2F.

    1.c. Table 2 does not show how many sorry theorems exists in each repository. For the Coxeter, I believe there are hundreds of sorrys, but paper only indicates the numbers 1 and 11, the number of sorry's it has been able to address without mentioning how many sorry have remained unproved. Table 2 has large margins of empty space, so I don't see why those numbers are not provided.

    1.d. Paper does not mention how many of those sorrys can be proved using native methods in Lean such as apply?, hint, or exact?.

     1.e. Many of the statements in the paper seem imprecise and perhaps trying to exaggerate the contributions of the paper. For example, the paper claims: “For the Coxeter repository, LeanAgent proves a complex lemma about Coxeter systems, showcasing its proficiency in group theory.” While the paper claims “proficiency” of its model in group theory, it is useful to note that its proof only has three lines of code, and the model fails to prove all other theorems in that repository. So the claim of “proficiency in group theory” seems overblown and unjustified.

2. Lifelong learning:

    2.a. The paper does not establish how its LLM suffers from the issue of forgetting when it comes to ATP. It seems that the paper has struggled with this, and has tried to find the best setting for training the LLM on multiple datasets in sequence, but it is not clear what the baseline was, what kind of accuracy was obtained, how the LLM was suffering from forgetting, and how the issue can be addressed.

    2.b. The paper keeps talking about the trade-offs between the stability and plasticity, but these two terms remain undefined and unquantified.

    2.c. It seems that, ultimately, paper has just performed trials and errors to eventually find the setting that yields the highest accuracy. Is that the case? I hope authors can clarify this, and explain their exact contribution. Is the contribution a method on how to train LLMs on multiple datasets while avoiding the issue of catastrophic forgetting, or is their contribution just an improvement on the task of ATP? If the former is the case, it would make sense to formalize the method, perform ablation studies, and demonstrate the generalization ability of the method. By ablation study, I mean reporting the accuracies in each configuration of lifelong learning and on each of the datasets. By accuracy for each dataset, I mean the number of proved theorems divided by the total number of theorems in a dataset.


Some other comments

The discussion about lifelong learning, and the claims about how mathematicians work seems like an incoherent opinion piece without much evidence, and not even necessary for the experiments of the paper.

The claim that “mathematicians often formalize across multiple domains and projects simultaneously or cyclically”, and giving Terry Tao as an example, seems a bit out of context. Terry Tao is famous for his unusual ability to work on multiple domains. Overall, I think the paper makes claims that are not even necessary. Paper can simply review the list of problems in miniF2F that it has failed to prove, perhaps showcase one of the more easier problems that it has not been able to prove, and be more grounded in reality when claiming “proficiency” in topics such as group theory.

**Questions:**

Please see weaknesses.

---

> ### Author Response · Authors · 2024-11-15
> **Response to Reviewer W5az**
>
> ## 1. I found the presentation of results very disorganized and rather unusual in the context of the ML literature. Overall, the presentation seems to be a stream of consciousness.
> __Answer:__ We thank the reviewer for this concern. We would highly appreciate it if you could point us to specific instances where we can improve our organization and limit the stream of consciousness. This would help us address your concern more precisely.
>
> ## 2. I tried for some time to find out what is the achieved accuracy of the proposed system on the miniF2F validation and test sets. I was shocked to see in the main body of the paper, in Tables 1 and 2, the paper provides the number of theorems in the datasets that it has proved without giving the total number of theorems nor the accuracy. I was expecting these tables to provide the accuracy of the proposed model on each of the datasets. I was only able to find somewhere deep in the appendix that the paper has been able to prove 99 theorems from the miniF2F dataset (not clear whether it is for the test set or valid set) which translates to accuracy of about 40%. It is as if the paper is trying to hide the accuracy on the known benchmark miniF2F. + The accuracy of 40% on miniF2F does not appear to be an improvement in the recent literature. For example, the accuracy of Hypertree Proof Search from 2022 is 41% on the miniF2F.
> __Answer:__ Thank you for the detailed feedback. First, we would like to emphasize that our intention was not to hide LeanAgent’s accuracy on MiniF2F. In Table 2, we provide the TPPS comparison between LeanAgent and ReProver on MiniF2F (225 vs. 86), demonstrating LeanAgent’s superior performance. As such, this functioned as a measure of accuracy for LeanAgent on MiniF2F. Please note that we have now removed the TPPS metric from the paper and have instead used a direct comparison with ReProver in terms of the total number of proven theorems. As such, Table 2 now displays 99 vs. 95 instead of 225 vs. 86.
>
> Moreover, we design LeanAgent to continuously generalize to and improve on ever-expanding mathematical knowledge without forgetting previously learned knowledge. Our goal is not to beat the MiniF2F benchmark; instead, we aim to perform well in proving *sorry* theorems across a range of diverse repositories. Other approaches, such as Hypertree Proof Search, focus on different objectives and don't address the lifelong learning aspect that is central to our work. As such, it is important to note that, in line with our existing setups for other repositories, we did not treat MiniF2F as a benchmark. Instead, we disregarded its existing splits and compared LeanAgent's proven *sorry* theorems overall with those of ReProver.
>
> However, we have now evaluated LeanAgent on the Lean4 version of the miniF2F test set using the same experimental setup from the paper, taking care not to progressively train on `Test.lean` and only proving *sorry* theorems from it. The results are as follows:
> - LeanAgent: Pass@1 of 38.1% (93 / 244 theorems)
> - ReProver (our run on Lean4) Pass@1 of 34.0% (83 / 244 theorems)
>
> Again, our goal is not to beat the MiniF2F benchmark, so we do not aim to report a Pass@1 that far outperforms other approaches. Please note that ReProver was only tested on the Lean3 test set in the LeanDojo paper, so we ran ReProver on the Lean4 test set for a fairer comparison. We have included these numbers in the paper, along with the reported success rates on the Lean4 test set from previous works.
>
> However, we would like to emphasize that LeanAgent is a lifelong learning framework, not a model. The performance of LeanAgent is dependent on the retriever used as the starting point. As such, although we now include the Pass@1 metrics for other methods besides ReProver in the paper, these metrics cannot be directly compared with LeanAgent. Such a comparison would be impractical for reasons including differences in data, pretraining, and fine-tuning. Again, we only compare with ReProver because we use ReProver’s retriever as the starting one in LeanAgent, allowing for a more faithful comparison. As mentioned in the introduction, we use ReProver because past work has shown that retrieval leads to improved generalizability.

---

> ### Author Response · Authors · 2024-11-15
> **Response to Reviewer W5az (continued)**
>
> ## 3. Table 2 does not show how many sorry theorems exists in each repository. For the Coxeter, I believe there are hundreds of sorrys, but paper only indicates the numbers 1 and 11, the number of sorry's it has been able to address without mentioning how many sorry have remained unproved. Table 2 has large margins of empty space, so I don't see why those numbers are not provided.
> __Answer:__ We thank the reviewer for bringing up this important point. Initially, we did not show the total number of *sorry* theorems in each repository in Table 2 because we received some feedback that it was confusing to show this number paired with the TPPS of LeanAgent and ReProver. However, we have now removed the TPPS metric from the paper and have instead used a direct comparison with ReProver in terms of the total number of proven theorems in Table 2. As such, we have included the total number of *sorry* theorems in each repository. For completeness, Coxeter only has 15 *sorry* theorems on the commit that we evaluated LeanAgent on (96af8aee7943ca8685ed1b00cc83a559ea389a97).
>
> ## 4. Paper does not mention how many of those sorrys can be proved using native methods in Lean such as apply?, hint, or exact?.
> __Answer:__ Thank you for bringing up this concern. We did not compare with native methods such as the ones listed because they are not ML techniques. The purpose of LeanAgent was to improve upon ML research for theorem proving, such as ReProver. Moreover, these methods are not frameworks, but ReProver was included as the starting point of the LeanAgent framework, which is why we compare LeanAgent to ReProver. Rather than comparing against existing tools, we aim to understand how lifelong learning can work in theorem proving. We will include this justification in Appendix A.2.
>
> ## 5. Many of the statements in the paper seem imprecise and perhaps trying to exaggerate the contributions of the paper. For example, the paper claims: “For the Coxeter repository, LeanAgent proves a complex lemma about Coxeter systems, showcasing its proficiency in group theory.” While the paper claims “proficiency” of its model in group theory, it is useful to note that its proof only has three lines of code, and the model fails to prove all other theorems in that repository. So the claim of “proficiency in group theory” seems overblown and unjustified.
> __Answer:__ We thank the reviewer for this valuable insight. We would like to emphasize that we do not aim to exaggerate the contributions of the paper. Our goal is to present our work and findings objectively. As such, we will update our statements to instead discuss stronger understanding relative to ReProver. For example, although this Coxeter proof is only three lines of code, ReProver could not generate this proof. As such, our statements now describe this stronger understanding of group theory relative to ReProver, aiming for precision. We will also update other instances of imprecise statements accordingly.

---

> ### Author Response · Authors · 2024-11-15
> **Response to Reviewer W5az (continued)**
>
> ## 6. The paper does not establish how its LLM suffers from the issue of forgetting when it comes to ATP. It seems that the paper has struggled with this, and has tried to find the best setting for training the LLM on multiple datasets in sequence, but it is not clear what the baseline was, what kind of accuracy was obtained, how the LLM was suffering from forgetting, and how the issue can be addressed.
> __Answer:__ Thank you for the detailed feedback. First, we would like to emphasize that LeanAgent is a lifelong learning framework, not an LLM. Moreover, although we use ReProver as a baseline for *sorry* theorem proving, no other lifelong learning frameworks for theorem proving exist in the literature. Since no lifelong learning baseline exists, we conduct an ablation study with seven lifelong learning metrics in Section 4.3 to showcase LeanAgent's superior handling of the stability-plasticity tradeoff, including the issue of forgetting. Specifically, the ablation study consists of seven additional setups constructed from a combination of learning and dataset options. Table 4 provides a thorough comparison of lifelong learning metrics across these setups. The rest of Section 4.3 presents a detailed analysis of the lifelong learning metrics, starting with the Single Repository and Merge All analysis before arriving at a comparative analysis and insights to conclude that LeanAgent is best suited for continuous generalizability and improvement. This is due to reasons like its very low forgetting scores (WF5, FM, CFR). This suggests that LeanAgent’s methodology addresses the problem of forgetting. Then, we use these results to help explain LeanAgent's superiority in *sorry* theorem proving performance.
>
> For additional details, Appendix A.4 thoroughly describes our choice of metrics, their unique interpretation in the Merge All strategy, and the effect of EWC on our results. Moreover, Appendix A.5’s analysis contains a *sorry* theorem proving comparison between setups in Table 9 and multiple instances where we reason about the performance between setups by using the lifelong learning analysis with our detailed metrics.
>
> ## 7. The paper keeps talking about the trade-offs between the stability and plasticity, but these two terms remain undefined and unquantified.
> __Answer:__ We thank the reviewer for their concern. Since LeanAgent is a novel lifelong learning framework for formal theorem proving, the concepts of stability and plasticity are crucial for our work. As such, we define these concepts in the introduction, stating that  plasticity is the ability to learn and adapt while stability is the ability to retain existing knowledge. We explain how increasing plasticity to learn new tasks efficiently can lead to overwriting previously learned information. Furthermore, we detail how conversely enhancing stability to preserve old knowledge may impair the ability to acquire new skills. Achieving the right balance is key to continuous generalizability in formal theorem proving.
>
> Moreover, Section 4.3 presents various metrics to quantify stability and plasticity and their tradeoff, including Windowed-Forgetting 5 (WF5), Forgetting Measure (FM), Catastrophic Forgetting Resilience (CFR), Windowed-Plasticity 5 (WP5), and Incremental Plasticity (IP). We define these metrics in Table 3 and describe in Section 4.3 why we introduce CFR and IP to address specific aspects of lifelong learning in theorem proving. Moreover, Appendix A.4 thoroughly describes this choice of metrics and their unique interpretation in the Merge All strategy.

---

> ### Author Response · Authors · 2024-11-15
> **Response to Reviewer W5az (continued)**
>
> ## 8. It seems that, ultimately, paper has just performed trials and errors to eventually find the setting that yields the highest accuracy. Is that the case? I hope authors can clarify this, and explain their exact contribution. Is the contribution a method on how to train LLMs on multiple datasets while avoiding the issue of catastrophic forgetting, or is their contribution just an improvement on the task of ATP? If the former is the case, it would make sense to formalize the method, perform ablation studies, and demonstrate the generalization ability of the method. By ablation study, I mean reporting the accuracies in each configuration of lifelong learning and on each of the datasets. By accuracy for each dataset, I mean the number of proved theorems divided by the total number of theorems in a dataset.
> __Answer:__ Thank you for your concern. We would like to emphasize that we do not just perform a comparison between options through trial and error to eventually find the setting that yields the highest accuracy. Instead, we present LeanAgent as a novel lifelong learning framework for formal theorem proving. It continuously generalizes to and improves on ever-expanding mathematical knowledge without forgetting previously learned knowledge. LeanAgent (1) derives complexity measures of theorems to compute a curriculum for learning, (2) progressively trains to learn while balancing stability and plasticity, and (3) searches for proofs of *sorry* theorems by leveraging a best-first tree search. LeanAgent uses a custom dynamic database to manage its evolving mathematical knowledge throughout this process. Please see the general response for more details. Again, we evaluate our framework against other possible setups through a thorough ablation study, rather than picking the best possible setup through trial and error. Moreover, our goal is not to beat ATP benchmarks such as MiniF2F. Instead, we aim to perform well in proving *sorry* theorems across a range of diverse repositories. The lifelong learning aspect is central to our work.
>
> Our sixth response to you contains a detailed response to the second part of your question.
>
> ## 9. The discussion about lifelong learning, and the claims about how mathematicians work seems like an incoherent opinion piece without much evidence, and not even necessary for the experiments of the paper. + The claim that “mathematicians often formalize across multiple domains and projects simultaneously or cyclically”, and giving Terry Tao as an example, seems a bit out of context. Terry Tao is famous for his unusual ability to work on multiple domains. Overall, I think the paper makes claims that are not even necessary. Paper can simply review the list of problems in miniF2F that it has failed to prove, perhaps showcase one of the more easier problems that it has not been able to prove, and be more grounded in reality when claiming “proficiency” in topics such as group theory.
> __Answer:__ We thank you for your detailed feedback. First, our seventh response to Reviewer SgJW contains lots of detail about the importance of lifelong learning in our work. Moreover, we use how mathematicians work as motivation for connecting mathematical knowledge between domains through continuous generalizability in lifelong learning. As evidence, we cite multiple Lean projects from Terence Tao and Patrick Massot. Again, we aim to use Terence Tao as context for our claims. We use him as an example because of his well-documented approach to formalizations in Lean, which is relatively rare among mathematicians in the field of ATP. Moreover, we heavily use lifelong learning, motivated by how mathematicians formalize, in our experiments. For more details, please see our sixth response to you. Furthermore, as mentioned previously, we have included the total number of *sorry* theorems in each repository in Table 2, showing the number of problems of MiniF2F that LeanAgent has failed to prove. In Appendix A.5, we detail 7 MiniF2F theorems that ReProver could prove but LeanAgent could not. As we mention in the paper, this is largely because ReProver is more tailored to simpler computation problems, which we corroborate through our analysis in that section. In addition, as mentioned previously, we would like to emphasize that we do not aim to exaggerate the contributions of the paper. We have updated our statements to instead discuss stronger understanding relative to ReProver.

---

> ### Author Response · Authors · 2024-11-23
> **Follow-up Response**
>
> We are writing to kindly remind you that we posted our response 7 days ago. If you have any additional feedback, concerns, or questions regarding our response, we would greatly appreciate hearing from you.

---

> > ### Comment · Reviewer_W5az · 2024-11-25
> >
> > I thank the authors for their detailed response, the improvements in the paper, and their engagement in the rebuttal.
> >
> > I am still not convinced about the composite metric and the way the paper assigns weights to the components in the metric. What authors want to measure can be expressed in terms of accuracy: number of proven theorems divided by the total number of theorems in a dataset. If we want to measure how a model is performing after being fine-tuned on a new dataset, we can still measure the accuracy of the model on both the old and new datasets. Still, I do not see a single reporting of the accuracy of the model in the main body of the paper. Departing from the accuracy metric isolates this paper from the literature. And the weights chosen for the newly proposed composite score seem to be completely random.
> >
> > If we were dealing with a paper using linear regression, we would have seen some basic sensitivity analysis on the chosen weights, and some justification that the metric should be linear. I could not find any justification for the chosen weights in the composite score. If everybody designs a new random metric for their own method without comparing the metric to the previously established metric, the community would not be able to make collective progress.
> >
> > Moreover, the comparison with ReProver does not seem to be clear, especially regarding the benefits of lifelong learning. Why is not a good idea to train the ReProver on all the data at once? As I requested in my original review, one would need to see the accuracy metric for all these settings in a clear ablation study.
> >
> > If we accept the argument of the paper that solely comparing with ReProver is enough, we have to note that ReProver comes with a specific dataset called LeanDojo Benchmark. It is not clear why there is no evaluation of LeanAgent on the LeanDojo benchmark.
> >
> > Overall, I think the revisions are improving the paper, but in my view, the current shortcomings put the paper below the borderline for acceptance. (This is how I view the paper. I hope my comments will be useful for the AC make a recommendation, and I hope authors will find the comments helpful, too.)

---

> > > ### Author Response · Authors · 2024-11-25
> > > **Follow-up Response**
> > >
> > > We thank the reviewer for their detailed response. We have removed the lifelong learning composite score from the paper and have instead used just the six separate lifelong learning metrics. Moreover, we now report the accuracy (proven theorems / total theorems) of LeanAgent and ReProver in Table 2 in the main body of the paper and Table 10 in the appendix.
> > >
> > > We are currently running an experiment to identify the result of finetuning ReProver on all the data without a curriculum. We will provide an update once the experiment is finished.
> > >
> > > We would like to emphasize that since ReProver is a ByT5 model finetuned on the LeanDojo Benchmark, and LeanAgent uses ReProver as the starting retriever, then LeanAgent does contain knowledge from the LeanDojo Benchmark. Evaluating on it would not provide meaningful insights about lifelong learning since it represents LeanAgent's starting knowledge, not new learning. In other words, we do not include the LeanDojo Benchmark as part of the curriculum because the benchmark is the first dataset LeanAgent has seen. Instead, we focus on evaluating LeanAgent's ability to continuously generalize to and improve on ever-expanding mathematical knowledge beyond its initial knowledge without forgetting previously learned knowledge.

---

> > > > ### Comment · Reviewer_W5az · 2024-11-27
> > > >
> > > > These changes all sound good to me. I look forward to seeing the result of the experiment, and will consider adjusting my score based on that.

---

> > > > > ### Author Response · Authors · 2024-11-27
> > > > > **Experimental Results**
> > > > >
> > > > > We thank the reviewer for their patience. We have finished running an experiment to identify the result of finetuning ReProver on all 23 repositories at once, called ReProver+. We have added the numbers to Table 2. We notice that LeanAgent outperforms ReProver+ on several repositories, emphasizing the importance of curriculum learning within our lifelong learning framework. Please note that, as expected, ReProver+ performs better than ReProver on multiple repositories. We welcome your thoughts on this matter.

---

> > > > > > ### Author Response · Authors · 2024-12-01
> > > > > > **Follow-up Response**
> > > > > >
> > > > > > We are writing to kindly remind you that we posted our response 4 days ago. If you have any additional feedback, concerns, or questions regarding our response, we would greatly appreciate hearing from you.

---

> > > > > ### Author Response · Authors · 2024-12-03
> > > > > **Gentle Follow-up**
> > > > >
> > > > > As today is the deadline for reviewer responses, we wanted to check one final time if you have any remaining feedback, concerns, or questions on our paper and our previous responses. We thank you for your time and consideration throughout the review process.

---

### Official Review · Reviewer_563Q · 2024-11-03

**Soundness:** 3
**Presentation:** 3
**Contribution:** 2
**Rating:** 5
**Confidence:** 4

**Summary:**

The paper introduces LeanAgent, a framework that augments LLM-guided formal theorem proving with lifelong learning and retrieval from a dynamic database. Evaluations on several real-world Lean GitHub repos and miniF2F demonstrate the advantage of the framework over a baseline with only retrieval from a static database (ReProver).

**Strengths:**

1. Demonstrates the utility of curriculum learning with retrieval from a dynamic database in the application of neural-guided formal theorem proving.
2. Evaluation on real-world repos shows real-world applicability of the framework (although this also has caveats – see Weakness 5 below).

**Weaknesses:**

1. The way the abstract describes the results is very misleading. For example, “LeanAgent successfully proves 162 theorems previously unproved by humans” is understood to mean that LeanAgent solved 162 unsolved problems in math, whereas in reality many of these just happened to not have their proofs filled in because the authors of the repositories hadn’t gotten to them yet, or the proofs are intentionally left blank because they are exercises from Lean tutorials/textbooks. For example, there are 0 = 1 placeholder theorems and theorems that simply involve expanding definitions (thus proven by just “refl”). The main metric of comparison with the baseline is also misleading, where the “11x” improvement actually refers to 3 repos where ReProver proved nothing and LeanAgent proved 1 theorem – see my next point.
2. The metric of comparison with the baseline (“TPPS”) appears to be designed to inflate the result of LeanAgent. The metric’s definition depends on the model/framework – already not a good sign. For LeanAgent it is (# ReProver Theorems Proved) + 10 * (# New Theorems Proved) + 1, and for the baseline ReProver it is (# ReProver Theorems Proved) + 1. Notably, the metric for LeanAgent is defined as the metric for the baseline plus a non-negative number, so by the very definition of the metric it is impossible for LeanAgent to perform worse than the baseline. Furthermore, the extra factor of 10 in the definition of the metric for LeanAgent (that doesn’t appear in the definition of the metric for the baseline) appears arbitrary and requires more justification than just saying it is “relatively modest considering the large difficulty gap between basic arithmetic and abstract algebra”.
3. The paper only showed results of the LeanAgent framework with the ReProver retriever and tactic generator. The paper would be stronger if it also showed consistent improvements from LeanAgent applied on top of well-known math LLMs such as Llemma and DeepSeek-Prover and/or other search strategies such as Hypertree Proof Search.
4. While the paper frequently emphasizes its curriculum learning strategy, it does not appear to make a technical contribution in this regard (simple rule of training for 1 epoch on each successive repository) and its application to formal math has already been explored (see, e.g., https://arxiv.org/abs/2202.01344).
5. The evaluation datasets (with the exception of miniF2F) appear to be various Lean GitHub repos with no guarantee of cleanliness. Indeed, the authors point out themselves that one of the repos has several “0 = 1” placeholder “sorry” theorems. Even among actual “sorry” theorems, there’s significant variation in difficulty, e.g., there are some proven by just “refl” and others that are more involved. This makes it more difficult to contextualize and interpret the results. (See also question 5 below for a suggestion to possibly improve this.)
6. In terms of presentation, there’s a decent amount of repetition (e.g. curriculum learning was explained an excessive number of times)

**Questions:**

(Questions 1-3 are about typos and wording issues)
1. Line 262: Missing word in “We sorry theorem proving performance…”?
2. Lines 227-228: “Before validation, we precompute embeddings for all premises in the corpus to ensure these embeddings are consistent with LeanAgent’s current state.” What does this mean?
3. Lines 318-324: Not sure what “formalize” means here, since if I understood the paper correctly, LeanAgent doesn’t do auto-formalization and MiniF2F is not an auto-formalization benchmark. It is also unclear to me what “in parallel” in line 323 means.
4. It would help the reader to show some basic statistics about each repo, namely, the number of sorry theorems in each repo, the number/percentage proven by LeanAgent, and the number/percentage proven by ReProver. Some statistics regarding the dynamic database would also help, such as the number of theorems/premises from each repo and their breakdown into difficulty categories.
5. (Related to the TPPS metric) It would be helpful to see a more informative metric than the TPPS metric (see criticisms above). For example, it would be ideal to have difficulty estimates for all the “sorry” theorems (based on e.g. ground truth proof length), and then for each repo, show (with e.g. a histogram) for each difficulty category, how many “sorry” theorems there are, how many were proven by ReProver, and how many by LeanAgent.
6. How is it ensured that the proofs do not involve circular reasoning? In other words, how did you make sure you’re not proving a sorry theorem A using another sorry theorem B while using sorry theorem A in the proof of sorry theorem B? (The Alternate PFR Commit case study on pages 30-31 seem to suggest that circular reasoning did occur, using 0 = 1 placeholder theorems to prove other 0 = 1 placeholder theorems.)
7. The coefficients in the definition of the Composite Score obtained (lines 429-431) seem arbitrary – how were they determined? How sensitive are the results to these coefficients?

---

> ### Author Response · Authors · 2024-11-15
> **Response to Reviewer 563Q**
>
> ## 1. The way the abstract describes the results is very misleading. For example, “LeanAgent successfully proves 162 theorems previously unproved by humans” is understood to mean that LeanAgent solved 162 unsolved problems in math, whereas in reality many of these just happened to not have their proofs filled in because the authors of the repositories hadn’t gotten to them yet, or the proofs are intentionally left blank because they are exercises from Lean tutorials/textbooks. For example, there are 0 = 1 placeholder theorems and theorems that simply involve expanding definitions (thus proven by just “refl”). The main metric of comparison with the baseline is also misleading, where the “11x” improvement actually refers to 3 repos where ReProver proved nothing and LeanAgent proved 1 theorem – see my next point.
> __Answer:__ We thank the reviewer for this valuable and detailed feedback. To aid communication, we have instead opted for using the following wording: “LeanAgent successfully proves 162 theorems previously unproved *formally* by humans.” We hope that by including “formally,” we emphasize that although these theorems are solved problems in math, they do not currently have formal proofs in the repositories we evaluate LeanAgent on.
>
> Crucially, we would like to highlight that we aim to describe how LeanAgent outperforms existing (static) ML methods in theorem proving with the help of lifelong learning. Some of these proofs may be trivial for mathematicians to formalize. However, by comparing with a previous ML model, ReProver, we emphasize that these proofs are non-trivial for ML. Again, we claim to provide an ML tool to mathematicians rather than achieve mathematician-level intelligence to solve unsolved problems.
>
> In practice, we expect LeanAgent to be used as a tool in existing Lean workflows. For example, one use case is leaving *sorry* theorems throughout the repository and asking LeanAgent to provide proofs of those theorems. This can greatly help speed up formalization projects and reduce rounds of code golf. Moreover, we find that many mathematicians have not yet adopted Lean. We hope that LeanAgent can help these mathematicians learn the language by assisting with tutorials. As such, we have included examples of repositories with *sorry* theorems, a few of which are tutorials or textbooks, to demonstrate these LeanAgent use cases.
>
> Moreover, we include much more detail in our revised paper to explain how LeanAgent found exploits within Lean’s type system. For example, Terence Tao noted that the 0 = 1 placeholder theorems in his PFR repository allowed for loopholes in the repositories that are being fixed through methods like introducing a `definition_wanted` keyword in Lean which functions similarly to `proof_wanted`. After the fixes, this exploit should no longer be an issue. In addition, we note in the paper how LeanAgent proved two theorems in an alternate commit of PFR with just the tactic rfl. However, Terence Tao later mentioned that these statements were about multiDist, another *sorry* theorem. Since any two instances of *sorry* are automatically equal by rfl, LeanAgent exploits this technicality to prove the two theorems. Again, we emphasize this exploit in the paper for transparency, and this should not be an issue going forward once fixes are incorporated into the repository. These examples highlight a critical weakness in our current approach and underscore the need for more robust and well-tested benchmarks. We hope that discovering these exploitable weaknesses with placeholder definitions and theorems can ensure that future benchmarks prevent the exploitation of technicalities.
>
> We thank the reviewer for their concern regarding using TPPS as the metric of comparison. We have removed the TPPS metric from the paper and have instead used a direct comparison with ReProver in terms of the total number of proven theorems. Please see the general response for more details.

---

> ### Author Response · Authors · 2024-11-15
> **Response to Reviewer 563Q (continued)**
>
> ## 2. The metric of comparison with the baseline (“TPPS”) appears to be designed to inflate the result of LeanAgent. The metric’s definition depends on the model/framework – already not a good sign. For LeanAgent it is (# ReProver Theorems Proved) + 10 * (# New Theorems Proved) + 1, and for the baseline ReProver it is (# ReProver Theorems Proved) + 1. Notably, the metric for LeanAgent is defined as the metric for the baseline plus a non-negative number, so by the very definition of the metric it is impossible for LeanAgent to perform worse than the baseline. Furthermore, the extra factor of 10 in the definition of the metric for LeanAgent (that doesn’t appear in the definition of the metric for the baseline) appears arbitrary and requires more justification than just saying it is “relatively modest considering the large difficulty gap between basic arithmetic and abstract algebra”.
> __Answer:__ We thank the reviewer for their concern. We have removed the TPPS metric from the paper and have instead used a direct comparison with ReProver in terms of the total number of proven theorems. Please see the general response for more details.
>
> ## 3. The paper only showed results of the LeanAgent framework with the ReProver retriever and tactic generator. The paper would be stronger if it also showed consistent improvements from LeanAgent applied on top of well-known math LLMs such as Llemma and DeepSeek-Prover and/or other search strategies such as Hypertree Proof Search.
> __Answer:__ Thank you for bringing this point up. We would like to highlight how the LeanAgent methodology is general to multiple LLMs; specifically, the key components of curriculum learning, progressive training, and the dynamic database are model-agnostic and are not tuned to any LLM. As noted in the paper, past work showed that retrieval leads to improved generalizability, which is why we use ReProver in our experiments.
>
> Although LeanAgent can work with other LLMs such as Llemma and DeepSeek-Prover, we would like to emphasize that using these LLMs in our work would require architectural modifications that go beyond the scope of our current work. For example, the 7B model DeepSeek-Prover as well as the 7B and 34B Llemma models are not retrieval-based. As such, rather than progressively training a retriever, we would progressively train the entire model. Although this may be feasible with methods such as Gradient Low-Rank Projection (GaLore) within the review period, we note that this would lead to fundamentally different usage than we currently demonstrate. Specifically, rather than using a best-first tree search approach as we do with ReProver’s retriever and tactic generator, we may instead need to generate the entire proof at once. This setup is quite dissimilar from our current evaluation, and so it may take considerable engineering efforts in areas such as updating hyperparameters. Overall, not only would it take some engineering efforts to complete these tasks within the review period, but these results may be too dissimilar from our current evaluation framework. However, we understand that the lack of applying LeanAgent to other math LLMs is a limitation of the current work, and we will identify this as an important future direction.
>
> Moreover, we would like to note that because LeanAgent does not claim to contribute a new search algorithm, it can be used with other search strategies such as Hypertree Proof Search (HTPS). However, the source code for HTPS was only released 4 months ago on GitHub, which is why we did not use it thus far. Moreover, the code has little documentation, so it would be non-trivial to use HTPS within our current LeanAgent implementation. However, we understand that the lack of using other search strategies within LeanAgent is another limitation of the current work, and we will identify this as an additional important future direction.

---

> > ### Comment · Reviewer_563Q · 2024-11-24
> > **Response to authors' rebuttal (2, 3)**
> >
> > I thank the authors' removal of the TPPS metric from the paper and their acknowledgment of the limitation in evaluation.

---

> ### Author Response · Authors · 2024-11-15
> **Response to Reviewer 563Q (continued)**
>
> ## 4. While the paper frequently emphasizes its curriculum learning strategy, it does not appear to make a technical contribution in this regard (simple rule of training for 1 epoch on each successive repository) and its application to formal math has already been explored (see, e.g., https://arxiv.org/abs/2202.01344).
> __Answer:__ We thank the reviewer for their concern. We would like to emphasize that the novelty of LeanAgent lies in providing a combined system of multiple individual subparts rather than implementing entirely new technical contributions across the board.
>
> Moreover, restricting progressive training to one epoch helps balance stability and plasticity. Although we recognize that our progressive training strategy is relatively straightforward, we would like to emphasize its effectiveness. Incorporating more sophisticated methods like Elastic Weight Consolidation (EWC), which uses the Fisher Information Matrix to constrain important weights for previous tasks, results in excessive plasticity. The uncontrolled plasticity is due to the inability of these methods to adapt parameter importance as theorem complexity increases. This forces rapid changes in parameters crucial for learning advanced concepts. Such methods fail to adapt to the evolving complexity of mathematical theorems, making them unsuitable for lifelong learning in theorem proving. However, our progressive training approach does not suffer from such problems. This can be seen in Table 4, where LeanAgent obtains the highest composite lifelong learning score of 94%. Moreover, Table 9 and Appendix A.5 describe the theorem proving capabilities of different setups in our ablation study. They again demonstrate the effectiveness of our progressive training strategy.
>
> Moreover, we would like to emphasize that we use a novel curriculum learning strategy by utilizing the structure of Lean proofs to learn on increasingly complex mathematical repositories. However, the prior work described has different contributions. Specifically, as noted in Section 2, the work created a synthetic inequality generator to produce a curriculum of statements of increasing difficulty. Here, difficulty was driven by depth of composition of inequalities and complexity of the input expressions to the inequalities. Then, this work uses this generated curriculum to study if expert iteration can solve these inequality statements. We note that this is fundamentally different from our approach, where we don’t synthetically generate the curriculum, restrict to inequalities, or use the generated curriculum as a benchmark. Rather, our curriculum learning strategy is general to existing mathematical repositories through our complexity measures of theorems. In addition, our goal is to use the curriculum for lifelong learning, focusing on the ability to continuously generalize to and improve on ever-expanding mathematical knowledge without forgetting previously learned knowledge.
>
> Our seventh response to Reviewer SgJW contains much more detail on the crucial importance of our curriculum learning approach.

---

> > ### Comment · Reviewer_563Q · 2024-11-25
> > **Response to Authors' rebuttal (4)**
> >
> > I thank the authors' clarification on the focus of the contribution of LeanAgent (novel combination of existing components), and the explanation of the effectiveness of the simple curriculum strategy compared to existing ones.
> >
> > I have read the authors' 7th response to Reviewer SgJW. I have a **follow-up question**: are there results comparing LeanAgent with the result of finetuning ReProver on all the data (without a curriculum)? Currently the comparison is between ReProver and ReProver finetuned on all the data (with a curriculum), so the fact that the latter outperforms the former is unsurprising as training/finetuning usually improves the performance of the model. A direct comparison between a curriculum vs. no curriculum would better illustrate the merits of curriculum learning.

---

> ### Author Response · Authors · 2024-11-15
> **Response to Reviewer 563Q (continued)**
>
> ## 5. The evaluation datasets (with the exception of miniF2F) appear to be various Lean GitHub repos with no guarantee of cleanliness. Indeed, the authors point out themselves that one of the repos has several “0 = 1” placeholder “sorry” theorems. Even among actual “sorry” theorems, there’s significant variation in difficulty, e.g., there are some proven by just “refl” and others that are more involved. This makes it more difficult to contextualize and interpret the results. + (Related to the TPPS metric) It would be helpful to see a more informative metric than the TPPS metric (see criticisms above). For example, it would be ideal to have difficulty estimates for all the “sorry” theorems (based on e.g. ground truth proof length), and then for each repo, show (with e.g. a histogram) for each difficulty category, how many “sorry” theorems there are, how many were proven by ReProver, and how many by LeanAgent.
> __Answer:__ We thank the reviewer for the detailed comments and suggestions. We would like to emphasize that our work is focused on lifelong learning under the harsh constraints of limited data (more details in Appendix A.7). Thus, as applicable to repositories like PFR, the metric can be susceptible to artifacts that artificially inflate performance. This underscores the need for more robust and well-tested benchmarks. We acknowledge that some of these Lean GitHub repos have weaknesses, and we hope that the exploits we have found within these repositories can ensure that future benchmarks prevent the exploitation of technicalities. Our response to your first question provides more detail about these exploits, including the “0 = 1” placeholder *sorry* theorems.
>
> As mentioned previously, we have removed the TPPS metric from the paper and have instead used a direct comparison with ReProver in terms of the total number of proven theorems. Please see the general response for more details.
>
> Regarding alternate metrics, as mentioned in the paper, there is no standardized metric for proof complexity. As such, we decided on $e^S$, where $S$ is the number of proof steps, for the curriculum (justification in the paper). However, a key challenge when comparing with ReProver is that generated proofs are sometimes quite short yet challenging to generate. For example, our pull request with our proof of `condRho_of_translate` in the PFR repository has been merged. This theorem is notably hard to prove; neither ReProver nor the other setups in Table 9 could prove it. In addition, Terence Tao mentioned how, despite the proof being only one line, the repository maintainers would likely not have found such a short proof easily. Thus, we find that using a metric with proof length for proving performance would not accurately reflect this difficulty.
>
> ## 6. Additional presentation and clarity issues.
> __Answer:__ Thank you for the feedback. We will make the following changes:
> 1. Thank you for catching the repetition. We will make the writing more concise.
> 2. We appreciate you catching the missing word. We added the word “compare”.
> 3. We thank the reviewer for catching the clarity issue in lines 227-228. These lines refer to the process of updating the premise embeddings before evaluating LeanAgent’s performance during progressive training. During progressive training, LeanAgent’s retriever, and therefore the embeddings it generates, are continuously updated. Thus, we need to ensure that at the end of the current progressive training run, all premises in the corpus have embeddings generated by LeanAgent’s current state. This will help properly evaluate LeanAgent’s validation performance. We will update this sentence for clarity.
> 4. Thank you for bringing up the wording concerns. You are correct, LeanAgent doesn’t do auto-formalization. We are now using “prove” for clarity. Moreover, “in parallel” is indeed unnecessary and unclear in this sentence, so we have now removed it.
>
> ## 7. It would help the reader to show some basic statistics about each repo, namely, the number of sorry theorems in each repo, the number/percentage proven by LeanAgent, and the number/percentage proven by ReProver. Some statistics regarding the dynamic database would also help, such as the number of theorems/premises from each repo and their breakdown into difficulty categories.
> __Answer:__ We thank the reviewer for this valuable suggestion. We will incorporate it.

---

> ### Author Response · Authors · 2024-11-15
> **Response to Reviewer 563Q (continued)**
>
> ## 8. How is it ensured that the proofs do not involve circular reasoning? In other words, how did you make sure you’re not proving a sorry theorem A using another sorry theorem B while using sorry theorem A in the proof of sorry theorem B? (The Alternate PFR Commit case study on pages 30-31 seem to suggest that circular reasoning did occur, using 0 = 1 placeholder theorems to prove other 0 = 1 placeholder theorems.)
> __Answer:__ Thank you for bringing up this important point. Such a circumstance, albeit relatively rare, is possible. However, we have found that this is an issue with Lean itself rather than LeanAgent. As noted in our first response to you, the exploits find loopholes within the Lean type system, such as with the 0 = 1 placeholder theorems and setting two instances of *sorry* equal by rfl. Efforts are currently in progress within these formalization projects to fix these issues, such as introducing a `definition_wanted` keyword in Lean which functions similarly to `proof_wanted`. We hope that discovering these exploitable weaknesses can ensure that the design of future benchmarks prevents such circular reasoning going forward.
>
> ## 9. The coefficients in the definition of the Composite Score obtained (lines 429-431) seem arbitrary – how were they determined? How sensitive are the results to these coefficients?
> __Answer:__ We thank the reviewer for this question. Our fourth response to Reviewer 5WDb contains a justification for how these coefficients were determined.

---

> > ### Comment · Reviewer_563Q · 2024-11-25
> > **Response to authors' rebuttal (8, 9)**
> >
> > 8. Thanks for the clarification on circular reasoning. I had in mind a simple preprocessing that first removes all the sorry statements from a given file, and adds them one at a time to the file for the agent to prove. I'd imagine that enforcing an order in which sorry statements are proven is a simple way to avoid circular reasoning. (Assuming forward references are rare, the order can just be the order in which the statements appear in the file.)
> >
> > 9. I have read the authors' 4th response to Reviewer 5WDb, and appreciate their qualitative justification of the chosen coefficients. However, I believe that when measuring success across multiple dimensions (improvement, plasticity, stability), it is most appropriate to list the metrics separately, and the user can consider different tradeoffs according to their use cases. Plotting the previous Pareto front and showing that your framework pushes it is, in my opinion, the best way to demonstrate your approach.
> >
> > The paper mentions "To our knowledge, there isn’t a widely established composite metric that provides a single stability-plasticity trade-off score with the first six metrics in Table 3." I suspect this is because any attempt to do so will inevitably draw the criticism regarding the weightings applied to the different aspects. Similarly, I haven't seen anyone compute a "composite score" that combines the performance of their model and its computational cost--they're always drawn on a 2D plot where the y-axis is the performance and the x-axis is the computational cost.
> >
> > As a result, my recommendation would be to either remove the composite score entirely, or replace it with a Pareto plot.

---

> > > ### Author Response · Authors · 2024-11-25
> > > **Follow-up Response**
> > >
> > > We thank the reviewer for their detailed and prompt response. We have revised the statement in the abstract to the following
> > >
> > > “LeanAgent successfully generates formal proofs for 155 theorems across 23 diverse Lean repositories where formal proofs were previously missing, many from advanced mathematics.”
> > >
> > > This new wording better reflects that we focus on filling theorem statements that are currently have no formal proofs, rather than claiming that these theorems were never proved by humans. We welcome your thoughts on this matter.
> > >
> > > In addition, we have updated the abstract and Table 2 to reflect that we proved 155 theorems that were not exploits. We also clarified that the 7 exploits existed in the alternate PFR commit, 5 being the 0 = 1 placeholder statements and the other 2 being setting two instances of *sorry* automatically equal by rfl.
> > >
> > > We are currently running an experiment to identify the result of finetuning ReProver on all the data without a curriculum. We will provide an update once the experiment is finished.
> > >
> > > We have removed the lifelong learning composite score from the paper and have instead used just the six separate lifelong learning metrics.

---

> > > > ### Comment · Reviewer_563Q · 2024-11-26
> > > >
> > > > The revised sentence looks good to me!
> > > >
> > > > The revisions of the abstract and Table 2 look good to me.
> > > >
> > > > Looking forward to the results of the experiment that finetunes without a curriculum!
> > > >
> > > > I appreciate the removal of the composite score from the paper.

---

> > > > > ### Author Response · Authors · 2024-11-27
> > > > > **Experimental Results**
> > > > >
> > > > > We thank the reviewer for their patience. We have finished running an experiment to identify the result of finetuning ReProver on all 23 repositories at once, called ReProver+. We have added the numbers to Table 2. We notice that LeanAgent outperforms ReProver+ on several repositories, emphasizing the importance of curriculum learning within our lifelong learning framework. Please note that, as expected, ReProver+ performs better than ReProver on multiple repositories. We welcome your thoughts on this matter.

---

> ### Author Response · Authors · 2024-11-23
> **Follow-up Response**
>
> We are writing to kindly remind you that we posted our response 7 days ago. If you have any additional feedback, concerns, or questions regarding our response, we would greatly appreciate hearing from you.

---

> ### Comment · Reviewer_563Q · 2024-11-23
> **Response to authors (1)**
>
> I thank the authors for their detailed response.
>
> I appreciate the revised wording in the abstract, but simply adding the word "formally" does not seem sufficient to accurately represent the truth. Many of the theorems were in fact previously proven formally by humans, such as those in Lean tutorials/textbooks.
> As an example, the paper uses the avigad/mathematics_in_lean_source repository at commit 5297e0, where solutions to exercises are often given in the same file (e.g., [this](https://github.com/avigad/mathematics_in_lean_source/blob/5297e0fb051367c48c0a084411853a576389ecf5/MIL/C03_Logic/S03_Negation.lean#L99) sorry theorem is proven in the same file a few lines below [here](https://github.com/avigad/mathematics_in_lean_source/blob/5297e0fb051367c48c0a084411853a576389ecf5/MIL/C03_Logic/S03_Negation.lean#L107)).
> As a result, I don't think the authors' claim
> > although these theorems are solved problems in math, they do not currently have formal proofs in the repositories we evaluate LeanAgent on
>
> is true.
>
> I appreciate the authors' clarification on the main claim of their paper and LeanAgent's use cases.
>
> I appreciate the clarification and transparents in regards to LeanAgent exploiting loopholes in Lean's type system. I believe the paper's quantitative results would be more convincing if these cases were removed and only theorems proven without these exploits are included.
>
> Removing the TPPS metric sounds good to me.

---

> > ### Author Response · Authors · 2024-11-24
> > **Follow-up Response to Reviewer 563Q**
> >
> > We thank the reviewer for their detailed suggestions. We would like to emphasize that ​​despite the solutions being sometimes included in the source file, the fact that LeanAgent and ReProver still struggle with some of these proofs demonstrates that sometimes (a) these proofs are still nontrivial and (b) they do not necessarily focus on the answer included in the training data. For example, the theorem you have linked is one that neither LeanAgent nor ReProver could prove. As such, we believe that including these solutions in the source file may not necessarily be an issue. This also explains why LeanAgent still cannot prove 8 *sorry* theorems in the Mathematics in Lean Source repository. We welcome your thoughts on this matter.
> >
> > Moreover, to aid transparency, our updated paper includes much more detail in our revised paper to explain how LeanAgent found exploits within Lean’s type system. We also state that the count of 162 theorems includes these exploits. We emphasize these exploits in the paper for transparency, and this should not be an issue going forward once fixes are incorporated into the repository. These examples are due to the weakness of the current state of repositories rather than a fundamental shortcoming of our ML approach. We hope that discovering these exploitable weaknesses with placeholder definitions and theorems can ensure that future benchmarks prevent the exploitation of technicalities.

---

> > > ### Comment · Reviewer_563Q · 2024-11-24
> > >
> > > I appreciate the authors' prompt response.
> > >
> > > I would like to clarify that my point about solutions being sometimes included in the source file was in reference to the following sentence from the updated abstract:
> > > > LeanAgent successfully proves 162 theorems previously unproved formally by humans across 23 diverse Lean repositories, many from advanced mathematics.
> > >
> > > I believe that the fact that solutions exist in the source file contradicts that statement in the abstract, and would recommend an alternative wording of the sentence.
> > >
> > > I appreciate the transparency regarding the fact that the 162 proven theorems include these exploits. However, I believe the transparency can be improved by not including these exploits in the count in the abstract, since otherwise the abstract misleadingly implies 162 theorems were "actually" proven by LeanAgent. I would find a number that doesn't include these exploits to be more representative of the theorem proving capabilities of LeanAgent and ReProver, and would recommend numbers that don't include these exploits also be reported in the new Table 2.
> > >
> > > I appreciate the removal of TPPS, opting for a more transparent metric that is the actual number of sorry theorems proven.

---

> ### Comment · Reviewer_563Q · 2024-11-30
> **Response to new experimental results**
>
> The new experimental results look good to me!
>
> I have revised my Soundness score to 3 and Overall score to 5 (borderline reject).
>
> * _Explanation of increased score:_
> 1. While the manuscript originally had several issues in its soundness (e.g., misleading abstract, misleading TPPS metric, arbitrary Composite Score lifelong learning metric, and lack of comparison with a direct fine-tuning baseline), I'm satisfied with the authors' resolution of these issues.
> 2. I also had the concern that the paper's contribution is not very novel given that curriculum learning in the context of theorem proving has been studied before, but I'm partially satisfied with the authors' response that they demonstrated a novel system (retrieval + curriculum learning) in a novel evaluation setting (real-world GitHub repos) that demonstrates real-world applicability.
> * _Explanation of not higher score:_
> 1. The new experimental results are relatively weak. While curriculum learning is intended to improve performance on hard tasks, the improvement over the direct fine-tuning baseline is only seen on easy repositories (with the exception of PFR), with no improvement on a few of the hardest repositories.
> 2. Furthermore, I say I am only partially satisfied in Point 2 above because I still perceive the contribution of the paper to be a little weak.

---

> > ### Author Response · Authors · 2024-12-01
> > **Thank You**
> >
> > We greatly thank the reviewer for their detailed review! We very much appreciate the comments and feedback. The reviewer’s time and effort in engaging with our work is very meaningful to us.

---

> ### Comment · Reviewer_563Q · 2024-12-03
> **Final summary**
>
> I thank the authors for participating in productive discussions about the paper. From the start to the end of the discussion period, I have increased my score from the original 3 (reject) to 5 (borderline reject). A summary of my reasons can be found in my [previous response ("Response to new experimental results")](https://openreview.net/forum?id=Uo4EHT4ZZ8&noteId=zVjO5b4jTL).
>
> Sincerely,
>
> Reviewer 563Q

---

### Official Review · Reviewer_5WDb · 2024-11-03

**Soundness:** 2
**Presentation:** 3
**Contribution:** 4
**Rating:** 6
**Confidence:** 4

**Summary:**

This paper proposes LeanAgent, a lifelong learning framework for formal theorem proving in Lean. LeanAgent alleviates the catastrophic forgetting issue in life-long learning of formal language provers via retrieval-augmented generation, with progressive training on the retriever.

Four main techniques are adopted to reduce the difficulty of training or proof search, including
  * **Curriculum learning** based on quantifiable complexity measured by the number of reasoning steps, allowing the model to learn from easy to difficult repositories
  * **Dynamic database management** that manages existing proofs and incorporates new proof steps generated by LeanAgent
  * **Progressive training** which limits the number of epochs to 1 to improve stability for mitigating forgetting, and choose the iteration with the highest validation recall on top-10 retrieved premises to improve the plasticity of absorbing new information.
  * **_sorry_ Theorem proving** via best-first search, similar to Dijkstra's algorithm in the graph, where each node/state is an incomplete proof, each edge is a proof step, and the path length is measured by the negative log-likelihood.

Experiments are conducted on two aspects, 1) comparing the quantity and quality of newly discovered proofs, and 2) ablation over different setups to further strike a balance between stability and plasticity. The main result shows that LeanAgent significantly outperforms ReProver and discovers 162 new theorems across 23 Lean repositories.

**Strengths:**

The main strength of the paper is its significance, especially for the Math community. A method capable of automatically discovering new theorems/proofs can lead to powerful tools to accelerate the development of mathematics and expedite relevant scientific fields such as physics and machine learning.

Although each proposed technique alone is not novel, showing the effectiveness of the combined systems is non-trivial, demonstrating the novelty of the proposed methods.

**Weaknesses:**

The main concern centers around its quality. The provided demos are impressive and great, but the code availability and adopted benchmarks are questionable, specifically,

  * **Code availability**: for LeanAgent to be truly accessible for mathematicians, the code and corresponding inference pipeline are necessary. I am wondering if the authors intend to release the code soon.
  * **Adopted benchmarks**: a standard metric for evaluating Lean-based systems is the success rate on the miniF2F test set. I am wondering if the authors can provide the number, along with comparisons to baselines other than retrieval-based methods. Just including the reported success rates in the previous literature would be fine, e.g. [1][2][3][4][5].

Other detailed comments are listed as follows,
  - [Clarity] lines 262-263, typo: "We sorry theorem theorem proving performance between..." -> "We compare sorry theorem proving performance between..."
  - [Clarity] lines 373-374: terms like "masters" may not be sufficiently rigorous and objective. It would be recommended to provide some numbers to improve the objectivity of the statement.
  - [Clarity] lines 366-387: Better provide redirections to the corresponding section in the Appendix, e.g. line 2050 of HairyBallDiff. This improves the objectivity of the paper.
  - [Clarity] line 437, Table 4: it is recommended to add up/down arrows in metrics to indicate whether the lower or higher values are better
  - [Clarity] Appendix section A.5: the figures for different theorems can be rescaled to make the fonts similar in size.
  - [Quality] lines 316, 430-431: several metrics adopted in the paper are computed based on heuristics, where further justification is encouraged to be added. For example, under the proposed metrics, the rank order of modern LLMs, o_1 > GPT-4o> ChatGPT, would provide evidence that the chosen coefficient is proper.

I am considering raising my score if those concerns can be properly addressed in the rebuttal.

### Reference

[1]: Jiang, Albert Qiaochu, et al. "Thor: Wielding hammers to integrate language models and automated theorem provers." Advances in Neural Information Processing Systems 35 (2022): 8360-8373.

[2]: Jiang, Albert Q., et al. "Draft, sketch, and prove: Guiding formal theorem provers with informal proofs." arXiv preprint arXiv:2210.12283 (2022).

[3]: Zhao, Xueliang, Wenda Li, and Lingpeng Kong. "Decomposing the enigma: Subgoal-based demonstration learning for formal theorem proving." arXiv preprint arXiv:2305.16366 (2023).

[4]: Wang, Ruida, et al. "Theoremllama: Transforming general-purpose llms into lean4 experts." arXiv preprint arXiv:2407.03203 (2024).

[5]: Xin, Huajian, et al. "DeepSeek-Prover-V1. 5: Harnessing Proof Assistant Feedback for Reinforcement Learning and Monte-Carlo Tree Search." arXiv preprint arXiv:2408.08152 (2024).

**Questions:**

* I am wondering if the code will be released, and when it is expected to be released.
* I am wondering if it is possible to report the miniF2F success rate of the proposed method, given this is the conventional benchmark for Lean-based methods.

---

> ### Author Response · Authors · 2024-11-15
> **Response to Reviewer 5WDb**
>
> ## 1. Code availability: for LeanAgent to be truly accessible for mathematicians, the code and corresponding inference pipeline are necessary. I am wondering if the authors intend to release the code soon.
> __Answer:__ We thank the reviewer for this important concern regarding code availability. We are currently working on cleaning and documenting the code, and we intend to open-source it shortly. The release will include the complete LeanAgent framework: the curriculum learning implementation, progressive training pipeline, dynamic database management system, and best-first search prover. It will also include the methodology and implementation details described in Appendix A.1 and Appendix A.2. The code will also provide modular components for specific use cases, such as just proving or just progressive training, to allow for an easy-to-use inference pipeline. We also aim to add examples of integrating LeanAgent with existing Lean repositories. This aids usability and further research.
>
> Moreover, we have issued pull requests to the respective repositories with the newly proven *sorry* theorems. These pull requests provide detailed information about exactly which theorems were proved and what the generated proof is. This demonstrates how LeanAgent can be integrated into mathematicians' existing repository workflows.
>
> ## 2. Adopted benchmarks: a standard metric for evaluating Lean-based systems is the success rate on the miniF2F test set. I am wondering if the authors can provide the number, along with comparisons to baselines other than retrieval-based methods. Just including the reported success rates in the previous literature would be fine, e.g. [1][2][3][4][5].
> __Answer:__ Thank you for the valuable suggestion. We have evaluated LeanAgent on the Lean4 version of the miniF2F test set using the same experimental setup from the paper, taking care not to progressively train on `Test.lean` and only proving *sorry* theorems from it. The results are as follows:
> - LeanAgent: Pass@1 of 38.1% (93 / 244 theorems)
> - ReProver (our run on Lean4) Pass@1 of 34.0% (83 / 244 theorems)
>
> Please note that ReProver was only tested on the Lean3 test set in the LeanDojo paper, so we ran ReProver on the Lean4 test set for a fairer comparison. We have included these numbers in the paper, along with the reported success rates on the Lean4 test set from previous works.
>
> However, we would like to emphasize that LeanAgent is a lifelong learning framework, not a model. The performance of LeanAgent is dependent on the retriever used as the starting point - ReProver’s retriever in our case. As such, although we include the Pass@1 metrics for other methods besides ReProver, these metrics cannot be directly compared with LeanAgent. Such a comparison would be impractical for reasons including differences in data, pretraining, and fine-tuning. Again, we only compare with ReProver because we use ReProver’s retriever as the starting one in LeanAgent, allowing for a more faithful comparison. As mentioned in the introduction, we use ReProver because past work has shown that retrieval leads to improved generalizability.
>
> Moreover, we design LeanAgent to continuously generalize to and improve on ever-expanding mathematical knowledge without forgetting previously learned knowledge. Our goal is not to beat the MiniF2F benchmark; instead, we aim to perform well in proving *sorry* theorems across a range of diverse repositories. The other approaches mentioned focus on different objectives and don't address the lifelong learning aspect that is central to our work.
>
> ## 3. Additional clarity comments.
> __Answer:__ Thank you for the very thorough and detailed notes about clarity. We will make the following changes:
> 1. Thank you for catching the typo! We will fix this.
> 2. We appreciate the feedback that terms such as “masters” are not objective. We will update all instances of similar phrases.
> 3. Thank you for the suggestion to provide redirections to the Appendix. We will update the paper accordingly.
> 4. We thank the reviewer for the suggestion to use arrows for clarity. We will incorporate this feedback.
> 5. Thank you for the suggestion about the font sizes. We will update the figure sizes accordingly.

---

> ### Author Response · Authors · 2024-11-15
> **Response to Reviewer 5WDb (continued)**
>
> ## 4. [Quality] lines 316, 430-431: several metrics adopted in the paper are computed based on heuristics, where further justification is encouraged to be added. For example, under the proposed metrics, the rank order of modern LLMs, o_1 > GPT-4o> ChatGPT, would provide evidence that the chosen coefficient is proper.
> __Answer:__ We thank the reviewer for their suggestion on quality. However, we are a bit unclear on how the rank order of modern LLMs can contribute to the choice of coefficient. Can you please clarify this?
>
> Please note that we have chosen to remove the TPPS, which was defined on line 316. Instead, we have used a direct comparison with ReProver in terms of the total number of proven theorems. Please see the general response for more details.
>
> We would like to clarify our choice of coefficients in the composite score defined on lines 430-431. We have provided a balance of stability (60%), improvement (20%), and plasticity (20%). We note that we weight stability with 60% because of its core functionality in continuous generalizability: maintaining knowledge of fundamental mathematical concepts is crucial for building up to more advanced knowledge over time. Our results validate this; in many instances, we see LeanAgent progress from simple to complex understanding, such as with SciLean. We give improvement and plasticity equal weighting because both are necessary to maintain generalizability while fitting the problem setup of continuous improvement in a theorem proving setting. However, it is important not to have excessive plasticity, another reason why a 20% weight is adopted. In addition, our ablation studies in Table 9 demonstrate better performance with controlled plasticity and competitive stability and improvement over time, further corroborating our choice of coefficients.

---

> ### Comment · Reviewer_5WDb · 2024-11-21
> **Thank you very much for the explanation**
>
> Thank you for the detailed explanation and corresponding modifications in the paper. I think this provides reasonable improvements to the paper's quality.
>
> ### 4. [Quality] lines 316, 430-431: several metrics adopted in the paper are computed based on heuristics, where further justification is encouraged to be added. For example, under the proposed metrics, the rank order of modern LLMs, o_1 > GPT-4o> ChatGPT, would provide evidence that the chosen coefficient is proper.
>
> > Regarding the questions related to "rank order of modern LLMs can contribute to the choice of coefficient", my main concern is centered around the specific choices of those weighting factors, such as $X$ in the TPPs, or the value of stability (60%)/improvement (20%)/plasticity (20%) in Composite Scores. The qualitative intuition of striking a balance between different properties of the model is reasonable, but from a quantitative perspective, I am not quite sure why these exact values X=10, or 60%/20%/20% are good choices for measuring the performance. Since the claim of **11x** improvements is made based on these metrics, I think it is quite important to offer evidence to support the soundness of the proposed metrics.
>
> > One possible evidence is to compare different models under the proposed metrics. If well-known strong models perform better than weak models, it would corroborate the effectiveness of the metric, and help support the performance claims for LeanAgent as a result.
>
> Hope this can help clarify the concern.

---

> > ### Author Response · Authors · 2024-11-23
> > **Thank you very much for the response!**
> >
> > We thank the reviewer for their response. We have removed the TPPS metric from the paper and have instead used a direct comparison with ReProver in terms of the total number of proven theorems. Please see the general response for more details.
> >
> > As for the choice of coefficients in the composite score, we would like to emphasize LeanAgent is a lifelong learning framework, not a model. The performance of LeanAgent is dependent on the retriever used as the starting point - ReProver’s retriever in our case. As such, we have not compared LeanAgent with separate LLMs, as doing so would be out of the scope of the current work. Again, we only compare with ReProver because we use ReProver’s retriever as the starting one in LeanAgent, allowing for a more faithful comparison. Moreover, the other approaches mentioned focus on different objectives and don't address the lifelong learning aspect that is central to our work.

---

> > > ### Author Response · Authors · 2024-12-01
> > > **Follow-up Response**
> > >
> > > We are writing to kindly remind you that we posted our response 9 days ago. If you have any additional feedback, concerns, or questions regarding our response, we would greatly appreciate hearing from you.

---

> ### Comment · Reviewer_5WDb · 2024-12-02
> **Thank you very much for the revision and further clarifications**
>
> I would like to thank the authors for the revision and further clarifications. As the controversial metrics have been removed, I think the major drawback of the paper has been addressed. So I am willing to increase my overall score from 5 to 6.
>
> Any additional comparisons with other baselines would be encouraged, which can further improve the quality of the paper. Also, I would recommend using radar charts to illustrate LeanAgent's superiority in different aspects. Hope this suggestion can be helpful.

---

> > ### Author Response · Authors · 2024-12-02
> > **Thank You**
> >
> > We greatly thank the reviewer for their detailed review! We very much appreciate the comments and feedback. The reviewer’s time and effort in engaging with our work is very meaningful to us.
> >
> > We thank the reviewer for their suggestion and will incorporate it into the main paper if it is accepted.

---

### Official Review · Reviewer_SgJW · 2024-11-10

**Soundness:** 2
**Presentation:** 2
**Contribution:** 2
**Rating:** 6
**Confidence:** 2

**Summary:**

The paper presents LeanAgent, a framework that aims to improve automated theorem proving through lifelong learning across multiple mathematical domains. The key contributions include a curriculum learning strategy, progressive training of retrieval models, and management of evolving mathematical knowledge. The paper claims significant improvements over baselines, particularly in proving "sorry" theorems previously unproved by humans.

**Strengths:**

- LeanAgent introduces an interesting approach to adapting theorem provers across evolving repositories. And it uses curriculum learning strategy provides a systematic way to handle increasing mathematical complexity and demonstrates potential for improving retrieval models through continuous training
- This paper provides a framework that could be practically useful for filling in "sorry" theorems and it demonstrates ability to work across diverse mathematical repositories, shows promise for developing maintainable theorem provers that can adapt to changes-
- This paper has detailed ablation studies examining different components

**Weaknesses:**

# Evaluation Methodology Concerns
- TPPS metric (with 10x multiplier) may artificially inflate improvements over baseline
- Unclear separation between "during" vs "after" lifelong learning results
- Combined statistics from multiple 10-minute runs may give unfair advantage over baselines
# Results Reproducibility Issues
- Some reported baseline numbers appear inconsistent with other evaluations (ReProver's results)
- Need clearer accounting of which theorems are proved and how
# Technical Limitations
- Only updates retriever model, not tactic generator
- Unclear why curriculum learning is necessary vs training on all data
- Long training times (4-9 days) may limit practical applicability

**Questions:**

1. Could you provide separate statistics for theorems proved during vs after lifelong learning?
2. Why not update both retriever and tactic generator during training?
3. Have you compared to simple pre-training on all available data?
4. How do you ensure reported improvements aren't just from running multiple model checkpoints?
5. What are the key computational requirements for practical deployment?

---

> ### Author Response · Authors · 2024-11-15
> **Response to Reviewer SgJW**
>
> ## 1. TPPS metric (with 10x multiplier) may artificially inflate improvements over baseline.
> __Answer:__ We thank the reviewer for their concern. We have removed the TPPS metric from the paper and have instead used a direct comparison with ReProver in terms of the total number of proven theorems. Please see the general response for more details.
>
> ## 2. Unclear separation between "during" vs "after" lifelong learning results.
> __Answer:__ We thank the reviewer for bringing this point up. We will revise Table 2 to make the progression clearer with the following three columns: theorems proven during lifelong learning, additional theorems proven after lifelong learning, and the total number of theorems proven (sum of the previous two columns).
>
> ## 3. Combined statistics from multiple 10-minute runs may give unfair advantage over baselines.
> __Answer:__ Thank you for raising this important concern about experimental fairness. First, we would like to emphasize that our goal is not to maximize the number of theorems proved through multiple 10-minute runs. Rather, we aim to track LeanAgent’s learning progression, which is fundamental to evaluating our lifelong learning system. This is why we evaluate our system both during and after lifelong learning. Table 2 now reports the theorems proved separately during and after lifelong learning to enable a clear comparison with the baseline at each stage.
>
> It is important to note that in practical deployment, it would not be necessary to run both 10-minute runs. The two-stage evaluation in our paper serves purely to demonstrate learning progression. Instead, normal usage would be much lighter, focusing on progressive training and proving theorems as the user desires. This would mean the user typically only needs one proving attempt per theorem.
>
> ## 4. Some reported baseline numbers appear inconsistent with other evaluations (ReProver's results).
> __Answer:__ We thank you for raising this point. We would highly appreciate it if you could point us to specific instances where these discrepancies appear. This would help us address your concern more precisely.
>
> Moreover, to address transparency about our baseline implementation, we would like to note that we used ReProver's publicly available retriever and tactic generator models from HuggingFace.
>
> ## 5. Need clearer accounting of which theorems are proved and how.
> __Answer:__ We thank the reviewer for their feedback. We have issued pull requests to the respective repositories with the newly proven *sorry* theorems. These pull requests provide detailed information about exactly which theorems were proved and what the generated proof is.
>
> ## 6. Only updates retriever model, not tactic generator
> __Answer:__ We thank the reviewer for bringing up this important point. We initially considered updating both models. However, we noticed that very few repositories we evaluate with introduce many new tactics. For example, SciLean does contain a `SciLean/Tactic` folder with source files to define new tactics, but we found this to be very rare. Since progressive training relies on more data than existed in pretraining to effectively update the model, the lack of many new tactics wouldn’t contribute to the tactic generator’s new knowledge. As such, we chose to update just the retriever model and not the tactic generator. This also aligns with prior work, such as LeanDojo, which states that premise selection is a critical bottleneck in formal theorem proving. Moreover, updating the tactic generator would add overhead to our framework, restricting its practical usage. As such, we avoid updating both models.

---

> ### Author Response · Authors · 2024-11-15
> **Response to Reviewer SgJW (continued)**
>
> ## 7. Unclear why curriculum learning is necessary vs training on all data.
> __Answer:__ Thank you for your feedback on the clarity of the problem setup. We would like to clarify the motivation behind using curriculum learning for lifelong learning in theorem proving. In the introduction, we note how mathematicians often formalize across multiple domains and projects simultaneously or cyclically. We use this as motivation for connecting mathematical knowledge between domains. Moreover, a key use case is formalizing new repositories without retraining on the entire dataset each time. When a mathematician creates some new repositories and adds them to a curriculum, they can simply progressively train LeanAgent on the new repositories while maintaining performance on previous ones, as shown in our experiments with both the initial curriculum and sub-curriculum. This is much more practical than retraining on the entire dataset, as that would be expensive in terms of compute and time, especially as the number of repositories grows.
>
> Moreover, data scarcity is a major problem in theorem proving, which we will describe even more in the revised version. As such, having enough high-quality data for effective pre-training on all repositories may not be feasible. We will also note how data scarcity in formal theorem proving hinders the generalizability of existing approaches. Training on all data, an approach similar to existing work, could prevent the model from generalizing to new repositories. However, LeanAgent does not have this restraint as it continuously generalizes to and improves on ever-expanding mathematical knowledge without forgetting previously learned knowledge.
>
> In addition, the results in Table 4 show that our lifelong learning setup leads to effective backward transfer, allowing learning on task A and then task B to improve performance on task A. This is a strong advantage that pre-training does not provide and is also why LeanAgent demonstrates progressive learning, starting with basic concepts and advancing to more complex ones. Also, although not a direct comparison to the pre-training approach, the "Merge All" strategy indicates decreased performance over time. This dataset strategy can be interpreted as being closer to pre-training than the “Single Repository” strategy, suggesting lower than desired pre-training performance when training on all data.
>
> Furthermore, Appendix A.6 explains the theory behind how curriculum learning is important in formal theorem proving. Prior work suggests that curriculum learning guides the learner towards better local minima in highly non-convex optimization landscapes such as that of theorem proving. This is an advantage that pretraining does not necessarily enjoy. Moreover, curriculum learning has been shown to have an effect similar to unsupervised pre-training, acting as a form of regularization. This allows LeanAgent to start from foundational to more complex knowledge in theorem proving. Moreover, the ability to act as a regularizer means curriculum learning prevents the model from overfitting to specific types of proofs or mathematical domains, an advantage that training on all data may not have. Moreover, multiple past works agree that curriculum learning provides larger gains when training data is limited. This is the case in formal theorem proving, where the number of formalized theorems and proofs is limited. These past works also state that curriculum learning leads to better generalization, supporting the observations of LeanAgent's superior lifelong learning metrics. Please see Appendix A.6 for more details.
>
> ## 8. Long training times (4-9 days) may limit practical applicability.
> __Answer:__ Thank you for the note on practical accessibility. Our entire experimental evaluation for the ablation study in Table 9 took 4 - 9 days on 4 A100 GPUs. However, this includes comprehensive experimental settings, such as validation, testing, and proving. Normal usage would be much lighter, focusing on just progressive training. For more details, please see our tenth response to you below.

---

> ### Author Response · Authors · 2024-11-15
> **Response to Reviewer SgJW (continued)**
>
> ## 9. How do you ensure reported improvements aren't just from running multiple model checkpoints?
> __Answer:__ Thank you for this important question. We would like to clarify how LeanAgent progressively trains its retriever on the newly generated dataset. We start with ReProver's retriever, a fine-tuned version of the ByT5 encoder. We then train LeanAgent on the new dataset for an additional epoch. Then, we save the model iteration with the highest validation recall for the top ten retrieved premises (R@10). We repeat this procedure for each dataset we generate from the database, hence the progressive nature of this training. As such, each progressive training iteration takes the previous checkpoint, updates it, and saves it as a new checkpoint. Crucially, this creates a single evolving model which builds on previous knowledge rather than multiple independent checkpoints. Each checkpoint is an updated version of the previous one, extending mathematical knowledge rather than starting from scratch. This strategy allows LeanAgent to continuously adapt to new mathematical knowledge from the premises in new datasets while preserving previously learned information. This is in contrast to repeatedly pre-training on all datasets, which starts from scratch each time to make different independent checkpoints.
>
> ## 10. What are the key computational requirements for practical deployment?
> __Answer:__ We thank the reviewer for this important question on practical use. LeanAgent is designed to be computationally efficient for practical deployment. For progressive training, each new repository only requires training for an additional epoch. Our current implementation uses 4 NVIDIA A100 GPUs for this progressive training because we can access this number of GPUs. However, recent advancements like Gradient Low-Rank Projection (GaLore) make progressive training increasingly accessible. GaLore enables training even 7B parameter models on consumer GPUs with just 24GB memory. Since ReProver’s retriever has less than 300 million parameters, progressive training can be quite computationally manageable.
>
> For inference, LeanAgent uses a best-first search strategy using relatively standard settings, such as generating 64 tactic candidates and retrieving 100 premises for each proof state with a 10-minute timeout per theorem. This can run on a single GPU, aiding accessibility to individual mathematicians. For storage, the dynamic database is lightweight, storing only metadata, theorems, proofs, and other traced data. A typical repository requires only up to a few hundred megabytes of storage.
>
> In practice, LeanAgent can be deployed in two ways: (a) Locally on a mathematician’s GPU for assistance with certain repositories or new commits, or (b) on a departmental/institutional server that periodically updates LeanAgent’s knowledge daily or weekly. This could involve updating with new commits across repositories from within the institution or with new repositories over time.
>
> Crucially, LeanAgent is much more efficient than training on all data: adding a new repository only requires up to a few more hours of training rather than expensive retraining on the complete dataset. As mentioned previously, our entire experimental evaluation for the ablation study in Table 9 took 4 - 9 days on 4 A100 GPUs. However, this includes comprehensive experimental settings, such as validation, testing, and proving. Normal usage would be much lighter, focusing on just progressive training.

---

> ### Author Response · Authors · 2024-11-23
> **Follow-up Response**
>
> We are writing to kindly remind you that we posted our response 7 days ago. If you have any additional feedback, concerns, or questions regarding our response, we would greatly appreciate hearing from you.

---

> > ### Author Response · Authors · 2024-12-01
> > **Follow-up Response**
> >
> > We are writing to kindly remind you that we posted our response 9 days ago. If you have any additional feedback, concerns, or questions regarding our response, we would greatly appreciate hearing from you.

---

> > > ### Comment · Reviewer_SgJW · 2024-12-02
> > >
> > > I would like to thank the authors for the revision. I think the most of my concerns has been addressed. I am willing to increase my overall score from 5 to 6.

---

> > > > ### Author Response · Authors · 2024-12-02
> > > > **Thank You**
> > > >
> > > > We greatly thank the reviewer for their detailed review! We very much appreciate the comments and feedback. The reviewer’s time and effort in engaging with our work is very meaningful to us.

---

### Author Response · Authors · 2024-11-15
**General Response**

We thank all of the reviewers for their very thoughtful and insightful feedback on our paper. We greatly appreciate the reviewers’ time and effort, as it will help us strengthen the presentation and clarity of our work.

Concerning multiple reviewers’ comments, we have removed the TPPS metric from the paper and have instead used a direct comparison with ReProver in terms of the total number of proven theorems.

For completeness, we want to emphasize that our intention was not to inflate the improvements over the baseline. There is no standardized measure of proof difficulty and proving performance, especially in a lifelong learning setting. As such, TPPS was an attempt towards this. We will add a detailed analysis of the reasoning behind and limitations of TPPS as an alternative metric to the number of proven theorems in Appendix A.7. What follows is some information regarding this.

Theorem proving difficulty is inherently non-linear in nature. For example, LeanAgent's significantly improved performance over the baseline across multiple repositories allows it to prove progressively harder theorems. Furthermore, *sorry* theorems lack ground truth proofs, so proving one is valuable. To address these nuances, one could propose the Theorem Proving Performance Score (TPPS) to emphasize newly proven *sorry* theorems.

However, it is important to recognize its preliminary nature and potential shortcomings. First, choosing the parameter $X$ is challenging. One approach is to choose a static value, such as $X = 10$, to standardize comparisons between LeanAgent and ReProver across diverse repositories. However, this may lead to inflated or deflated metrics on such repositories. Alternatively, $X$ could be chosen adaptively based on the difficulty of a repository, but this may similarly result in unrealistic metric scores. Overall, the TPPS metric focuses on quantity and faces challenges in ensuring comparisons across diverse repositories.

Furthermore, TPPS does not account for the difficulty of the theorems being proved. While we initially considered using proof length as a proxy for difficulty (e.g., $e^S$ where $S$ is the number of proof steps), we found this approach problematic during proving evaluation. LeanAgent sometimes generates shorter proofs for difficult theorems, making proof length an unreliable indicator of theorem difficulty.

In light of these considerations, we acknowledge that TPPS, while a step towards quantifying theorem-proving performance in a lifelong learning setting, has many limitations. Future work should focus on developing more sophisticated and robust evaluation frameworks that address these challenges. By highlighting these issues, we hope to contribute to the ongoing discussion on how best to evaluate and compare theorem-proving systems, particularly in the context of lifelong learning.


Overall, in this paper, we introduced LeanAgent, a novel lifelong learning framework for formal theorem proving. It continuously generalizes to and improves on ever-expanding mathematical knowledge without forgetting previously learned knowledge. The key contributions of our work are as follows:
1. LeanAgent combines curriculum learning based on proof complexity, progressive training to balance stability and plasticity, and best-first search for proving *sorry* theorems, all while using a dynamic database for mathematical knowledge.
2. LeanAgent successfully proves 162 *sorry* theorems across 23 diverse Lean repositories, demonstrating progression from basic to advanced mathematics. Moreover, LeanAgent earns a composite lifelong learning score of 94%, explaining its superior *sorry* theorem proving performance.

Recent research has explored using LLMs to generate proof steps or complete proofs for formal theorem proving. However, existing approaches typically train or fine-tune LLMs on specific datasets, and data scarcity hinders the generalizability of these approaches. Our work with LeanAgent addresses this limitation, paving the way for advanced AI tools to help mathematicians construct formal proofs. We believe that our work represents an essential step towards generalizable and democratized formal theorem proving, enabling researchers and mathematicians to prove cutting-edge theorems.

---

> ### Comment · Reviewer_563Q · 2024-11-23
> **Response about TPPS**
>
> I appreciate that the authors will be removing the misleading TPPS metric and replace the results table with the more transparent "# sorry theorems proven" out of total # of sorry theorems. I would appreciate it if the authors can show the revised table.

---

### Author Response · Authors · 2024-11-24
**Updated Paper**

We thank the reviewers for their responses. We have updated the paper with the suggested revisions and content from previous responses. We welcome any follow-up discussions!

---

### Meta-Review · Area_Chair_1yVa · 2024-12-19

**Metareview:**

The paper presents LeanAgent, a lifelong learning framework for formal theorem proving in Lean. LeanAgent integrates curriculum learning, retrieval-augmented generation, and progressive training to address catastrophic forgetting and improve continuous generalization across multiple mathematical domains. The framework demonstrates improvements over the baseline ReProver by proving sorry theorems across 23 diverse Lean repositories.

**Strengths**

* The paper tackles a relevant and challenging problem in theorem proving.
* The combination of curriculum learning, retrieval, and LLM-guided formal proof search offers a novel approach to lifelong learning.
* The experiments are comprehensive, with ablation studies validating each component.
* The authors made significant revisions addressing key concerns, including the removal of the TPPS metric and clarifying misleading claims.

**Weaknesses**

* Some concerns about novelty, as individual techniques (e.g., curriculum learning) have been explored before.
* Evaluation robustness could be improved with broader baseline comparisons (e.g., Hypertree Proof Search).
* Improvements are modest on harder tasks, raising questions about scalability.

The authors’ revisions and additional experiments have addressed most concerns. While some reservations remain, the overall consensus leans towards acceptance. The paper provides a meaningful contribution to lifelong learning in formal theorem proving, and the revisions have significantly improved clarity and transparency.

**Additional Comments On Reviewer Discussion:**

Post-rebuttal, reviewers appreciated the revisions, particularly the removal of the TPPS metric, clearer claims, and the ReProver+ comparison. While concerns about novelty and evaluation robustness remain, the consensus leans towards acceptance. For the camera-ready version, the authors should ensure transparent reporting, consider broader baselines, and further clarify evaluations on harder tasks.

---

### Decision · Program_Chairs · 2025-01-22

Accept (Poster)